# Dopamine D2 receptor regulates cortical synaptic pruning in rodents

Ya-Qiang Zhang[1,3], Wei-Peng Lin [1,2,3], Li-Ping Huang[1,3], Bing Zhao[1], Cheng-Cheng Zhang[1] & Dong-Min Yin [1✉]

Synaptic pruning during adolescence is important for appropriate neurodevelopment and synaptic plasticity. Aberrant synaptic pruning may underlie a variety of brain disorders such as schizophrenia, autism and anxiety. Dopamine D2 receptor (Drd2) is associated with several neuropsychiatric diseases and is the target of some antipsychotic drugs. Here we generate self-reporting Drd2 heterozygous (SR-Drd2$^{+/-}$) rats to simultaneously visualize Drd2-positive neurons and downregulate Drd2 expression. Time course studies on the developing anterior cingulate cortex (ACC) from control and SR-Drd2$^{+/-}$ rats reveal important roles of Drd2 in regulating synaptic pruning rather than synapse formation. Drd2 also regulates LTD, a form of synaptic plasticity which includes some similar cellular/bio-chemical processes as synaptic pruning. We further demonstrate that Drd2 regulates synaptic pruning via cell-autonomous mechanisms involving activation of mTOR signaling. Deficits of Drd2-mediated synaptic pruning in the ACC during adolescence lead to hyper-glutamatergic function and anxiety-like behaviors in adulthood. Taken together, our results demonstrate important roles of Drd2 in cortical synaptic pruning.

[1] Key Laboratory of Brain Functional Genomics, Ministry of Education and Shanghai, School of Life Science, East China Normal University, 200062 Shanghai, China. [2] Joint Translational Science and Technology Research Institute, East China Normal University, 200062 Shanghai, China. [3] These authors contributed equally: Ya-Qiang Zhang, Wei-Peng Lin, Li-Ping Huang. ✉email: dmyin@brain.ecnu.edu.cn

During the postnatal development, functionally important synapses are strengthened and stabilized, whereas unnecessary connections are weakened and eventually eliminated[1,2]. The process of synaptic pruning depends on neural activity during the limited postnatal period known as the 'critical time window'[3–6]. Proper synaptic elimination is essential to form a healthy and adaptive brain, while aberrant synaptic pruning may cause a variety of brain disorders including schizophrenia, autism, and anxiety[7–11]. Recent studies demonstrate that synaptic pruning is regulated by microglia-derived complement proteins and astrocyte-mediated phagocytosis[12–17]. However, the neuronal mechanisms that specifically regulate synaptic pruning is relatively less well understood.

Dopamine signaling plays important roles in neural development and brain functions such as locomotion, emotion, learning, and memory[18–20]. The brain contains two types of dopamine receptors based on sequence homology and function: the excitatory D1-like receptors (Drd1 and Drd5) and inhibitory D2-like receptors (Drd2, Drd3, and Drd4). Drd2 is genetically linked to several neuropsychiatric diseases and is the target of antipsychotics[21–26]. Studies from rats showed that dopamine innervation and Drd2 expression in the frontal cortex increased during adolescence and reached a peak in young adulthood[27–29]. Drd2, a $G\alpha_{i/o}$-coupled receptor, can inhibit cAMP signaling but activate β-arrestin and protein phosphatase 2 A (PP2A)[30]. Both cAMP inhibition and PP2A activation have been implicated in long-term depotentiation (LTD)[31], a form of synaptic plasticity which includes some similar cellular/biochemical processes as synaptic pruning[32].

Cortical neurons are heterogenous in gene expression and connectivity patterns[33]. Recent studies suggest that cortical neurons expressing Drd1 and Drd2 are generally segregated and have different subcortical projections[34,35]. The heterogeneity of cortical neurons makes it challenging to use regular genetic approaches to study the mechanisms of synapse development in different population of neurons. To address this issue, we generated self-reporting Drd2 heterozygous (abbreviated to SR-Drd2$^{+/-}$) rats to simultaneously visualize Drd2-positive neurons and downregulate Drd2 expression. We then used the SR-Drd2$^{+/-}$ rats to investigate the roles of Drd2 in synapse development in anterior cingulate cortex (ACC), a brain region implicated in emotional symptoms of neuropsychiatric diseases such as anxiety and depression[36,37].

Here we focus on dendritic spines where the majority of excitatory synapses are formed. We found that Drd2 specifically regulated dendritic spine pruning but not formation in the pyramidal neurons from deep layers of ACC. Moreover, Drd2 also regulates LTD, a form of synaptic plasticity which includes some similar cellular/biochemical processes as synaptic pruning. We further demonstrate that Drd2 regulates synaptic pruning through cell-autonomous mechanisms involving activation of mTOR signaling in sensitive periods. Lastly, inhibition of Drd2 in the ACC during adolescence lead to hyperglutamatergic function and anxiety-like behaviors in adulthood. Taken together, our results reveal mechanisms underlying synaptic pruning and how the deficits of these processes cause brain dysfunction in adulthood.

## Results

**Cellular expression pattern of Drd2 in rat ACC.** To explore the cellular expression pattern of Drd2 in rat ACC, we generated self-reporting Drd2 (abbreviated to SR-Drd2) rats by crossing Drd2-Cre with Cre-dependent tdTomato reporter (Ai14) rats (Fig. 1a). The Drd2-positive cells express tdTomato in the SR-Drd2 rats[38]. The slices of ACC from SR-Drd2 rats were collected and analyzed for tdTomato expression (Fig. 1b). We found very little Drd2-positive cells in the layer 1 of ACC from adult (2-month-old) SR-Drd2 rats (Fig. 1c, d). More Drd2-positive cells were distributed in the layer 5 than layers 2–3 of ACC (Fig. 1c, d). tdTomato was colocalized with NeuN (a marker for neurons), but not GAD67 (a maker for GABAergic interneurons) in layers 2–3 of ACC (Fig. 1e), indicating expression of Drd2 in putative pyramidal neurons. All the Drd2-positive cells were neurons while around 45% of neurons express Drd2 in the layer 5 of ACC (Fig. 1f, g). About 10% of Drd2-positve neurons were GABAergic interneurons in the layer 5 of ACC (Fig.1f, g). This notion was further supported by the results of electrophysiological recording that only 9% of Drd2-positive cells are fast-spiking GABAergic interneurons (Fig. 1h, i). Since interneurons represent on average around 20% of the cortical neurons, it is reasonable that Drd2-positive interneurons represent only 9–10% of the total, which is still almost half of the interneurons (Fig. 1g). Taken together, these results suggest that Drd2 is expressed in both pyramidal neurons and GABAergic interneurons in the deep layers of ACC.

To rule out the possibility of transient embryonic Cre expression resulting in false positive tdTomato-labeling, we injected Cre-dependent tdTomato reporter viruses into the rostral part of ACC from 5-week-old Drd2-Cre rats (Supplementary Fig. 1a–c). Three weeks after the injection, the ACC slices were collected and analyzed for tdTomato expression. The Drd2-positve cells were all neurons and were more distributed in layers 5–6 than layers 2–3 (Supplementary Fig. 1d–f). The results from virus injection indicate that the Drd2 is mainly expressed in neurons and in the deep layers of ACC from 5-week-old rats, which is consistent with the data from SR-Drd2 rats (Fig. 1). In the following study, we focused on the pruning of dendritic spines which were from pyramidal neurons in the layer 5 of ACC.

**Generation and characterization of SR-Drd2$^{+/-}$ rats.** To simultaneously label Drd2-positive neurons and downregulate Drd2 expression, we crossed the SR-Drd2 with Drd2$^{+/-}$ rats to obtain the SR-Drd2$^{+/-}$ rats (i.e., Drd2$^{Cre/-}$; Rosa26$^{td/+}$ rats) (Fig. 2a, b). The DRD2 protein levels in the ACC of SR-Drd2$^{+/-}$ rats were reduced by half compared with the SR-Drd2 rats, i.e., control rats (Fig. 2c). By contrast, the DRD1 protein levels were similar between control and SR-Drd2$^{+/-}$ rats (Fig. 2d). Both the body and brain weight were comparable between 2-moth-old control and SR-Drd2$^{+/-}$ rats (Supplementary Fig. 2a–d), indicating no growth retardation by the Drd2$^{+/-}$ mutation. The total neuron number and the number of Drd2-positive neurons did not change in the different layers of ACC from SR-Drd2$^{+/-}$ rats (Supplementary Fig. 2e–h), which suggest that Drd2$^{+/-}$ mutation did not affect the cortical lamination.

We next study whether Drd2$^{+/-}$ mutation altered dendritic arborization of Drd2-positive neurons in the layer 5 of ACC. To this end, we filled Drd2-positive neurons with biocytin and used immunofluorescence to reveal the dendritic morphology (Supplementary Fig. 3a). The extent of dendritic arborization was quantified by Sholl analysis (Supplementary Fig. 3b)[39]. The arborization of apical and basal dendrites of Drd2-positive neurons in the layer 5 of ACC is similar between control and SR-Drd2$^{+/-}$ rats (Supplementary Fig. 3c–f), which suggest that Drd2$^{+/-}$ mutation may not affect dendritic arborization.

**Regulation of dendritic spine pruning but not formation by Drd2.** Next, we investigate whether Drd2$^{+/-}$ mutation influenced dendritic spine growth of Drd2-positive neurons in the layer 5 of ACC. Toward this goal, we filled Drd2-positive neurons with biocytin and used immunofluorescence to reveal the dendritic

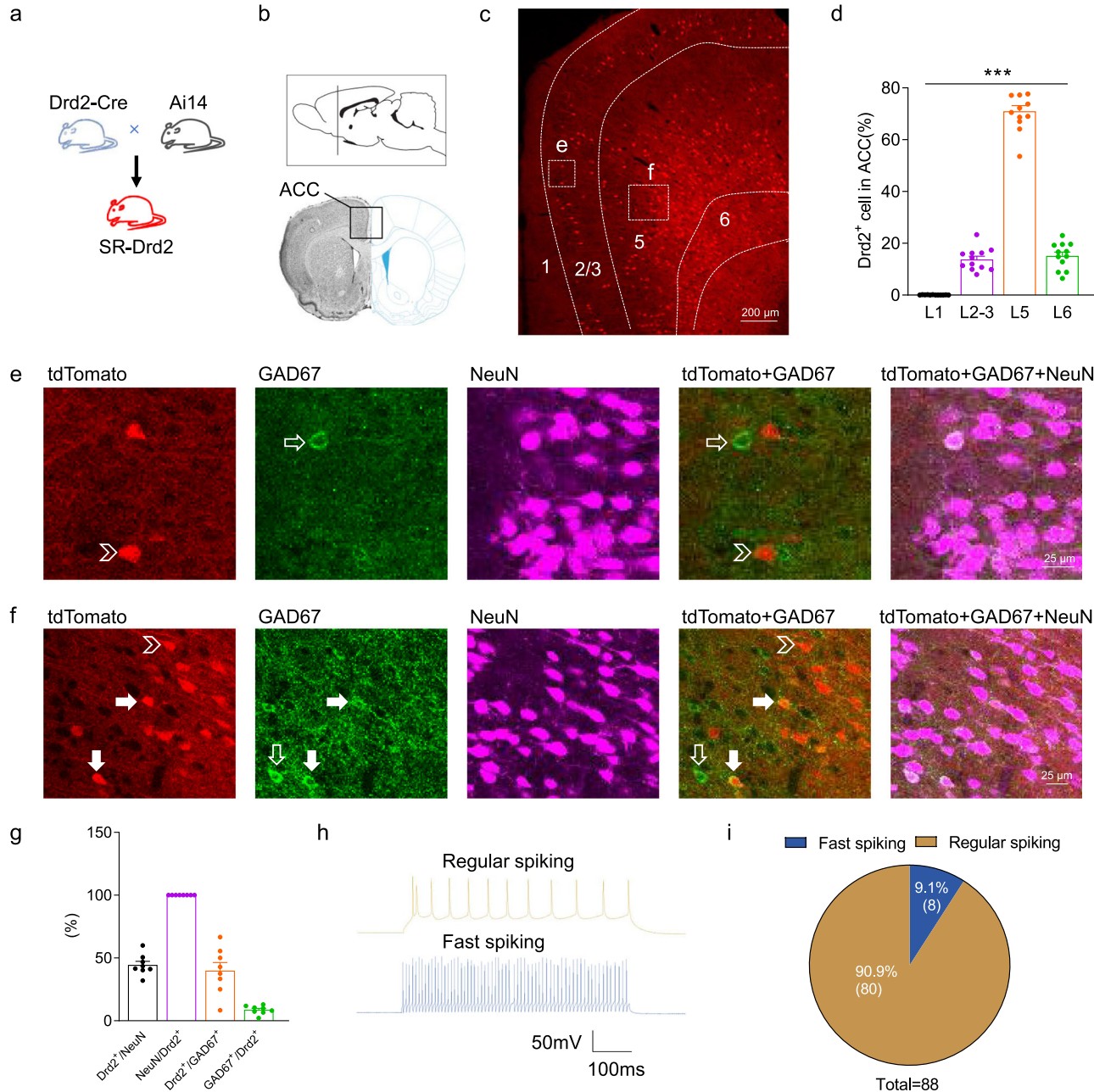

**Fig. 1 Cellular expression pattern of Drd2 in rat ACC. a** Generation of SR-Drd2 rats by crossing heterozygous Drd2-Cre rats with heterozygous Ai14 (Rosa26-LSL-tdTomato) rats. **b** Diagram of rat brain sagittal section (top) and coronal section (bottom). The dashed line of the sagittal section diagram indicates the position of the coronal section. The rectangle indicates the brain regions of ACC. **c** Expression of tdTomato in the ACC from SR-Drd2 rats. Scale bar, 200 μm. **d** The percentage of Drd2-positive cells in different layers of ACC among total Drd2-positive cells in ACC. ***$P < 0.0001$, one-way ANOVA, $n = 12$, Data are presented as mean values ± SEM. **e, f** Immunofluorescent images of tdTomato, GAD67, and NeuN from the rectangles in panel **c**. Arrowheads indicate Drd2-positive but Gad67-negative cells, empty arrows indicate Gad67-positive but Drd2-negative cells, solid arrows indicate Drd2-positive and Gad67-positive cells. Scale bar, 25 μm. **g** The percentage of neurons expressing Drd2 (Drd2+/NeuN = 44.47%). All Drd2-positive cells are neurons (NeuN/Drd2+ = 100%). The percentage of GABAergic interneurons expressing Drd2 (Drd2+/Gad67+ = 39.91%). The percentage of Drd2-positive neurons that were GABAergic interneurons (Gad67+/ Drd2+ = 8.693%). $n = 8$, Data are presented as mean values ± SEM. **h** Representative action potentials of regular spiking and fast-spiking neurons. **i** 90.9% of Drd2+ neurons are regular spiking neurons. $n = 88$ neurons from 10 SR-Drd2 rats.

spines. Here we focused on the secondary basal dendrites (Fig. 2e) where the majority of spines are formed on the layer 5 pyramidal neurons in cortex[40]. The spine densities of Drd2-positive neurons increased from 2 to 4-week-old age and then reduced toward 8-week-old age in control rats (Fig. 2f–h). These results indicate a synapse formation during early postnatal period followed by a net synaptic pruning during adolescence in Drd2-positive neurons.

The spine densities of Drd2-positive neurons from 2 to 4-week-old age were similar between control and SR-Drd2$^{+/−}$ rats (Fig. 2f–h), which suggest that Drd2$^{+/−}$ mutation may not affect synapse formation. However, the spine densities of Drd2-positive neurons were not reduced from 4- to 8-week-old age in SR-Drd2$^{+/−}$ rats (Fig. 2f–h), which indicate that Drd2$^{+/−}$ mutation may impair synaptic pruning. The deficits of synaptic pruning

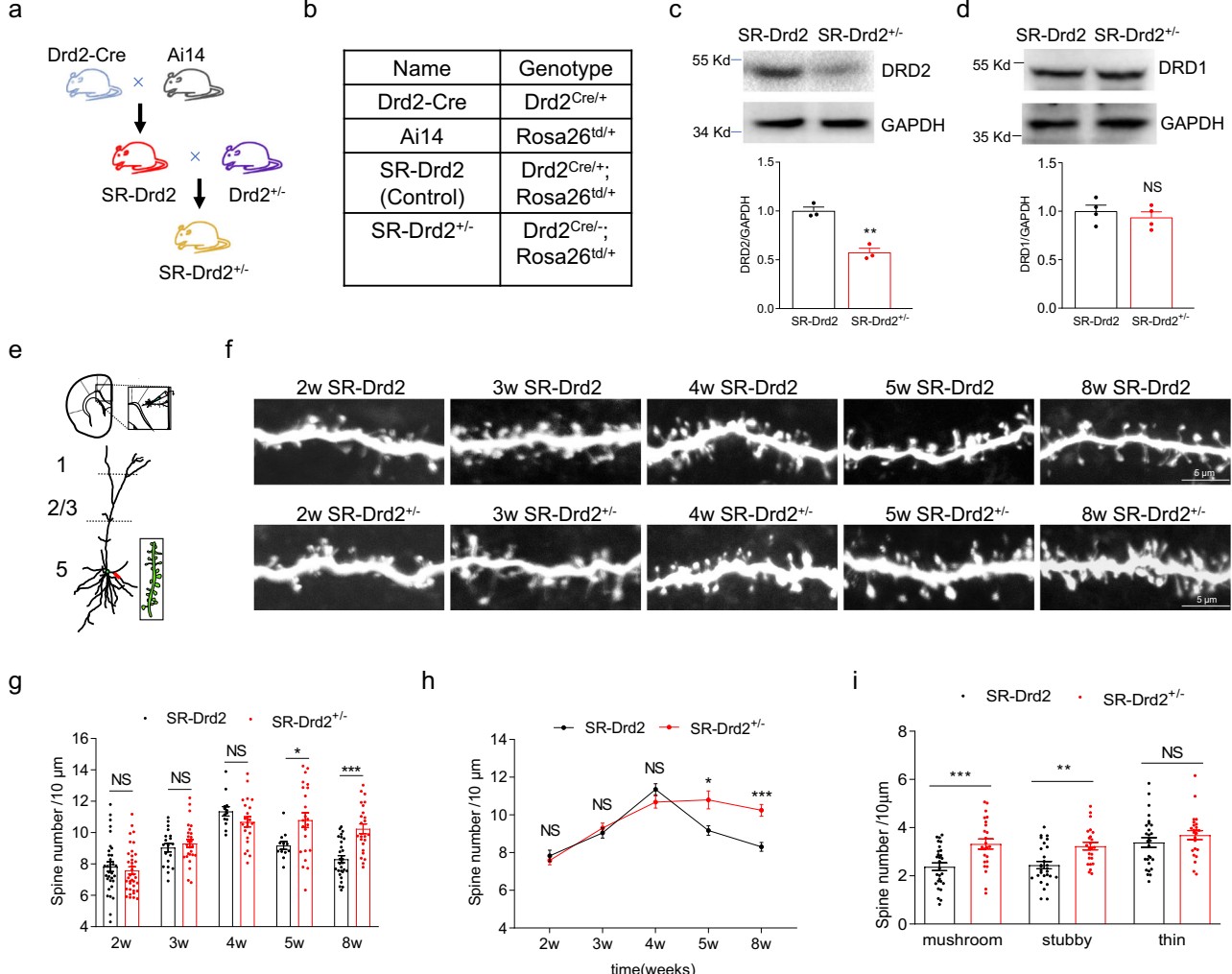

**Fig. 2 Regulation of dendritic spine pruning but not formation by Drd2. a** The strategies to generate SR-Drd2 (control) and SR-Drd2$^{+/-}$ rats. The SR-Drd2 rats were generated by crossing Drd2-Cre rats with Ai14 (Rosa26-LSL-tdTomato) rats. The SR-Drd2$^{+/-}$ rats were generated by crossing SR-Drd2 rats with Drd2$^{+/-}$ rats. **b** The genotypes of different lines of rats. **c** Decreased DRD2 protein levels in the ACC of SR-Drd2$^{+/-}$ rats. The homogenates of ACC from SR-Drd2 and SR-Drd2$^{+/-}$ rats were subjected to western blot and probed with anti-DRD2 and anti-GAPDH antibodies (top). Quantification of DRD2/GAPDH (bottom). **P = 0.0022, two-sided $t$-test, $n = 3$ for each group, data were normalized to controls. Data are presented as mean values ± SEM. **d** Unchanged DRD1 protein levels in the ACC of SR-Drd2$^{+/-}$ rats. The homogenates of ACC from control and SR-Drd2$^{+/-}$ rats were subjected to western blot and probed with anti-DRD1 and anti-GAPDH antibodies (top). Quantification of DRD1/GAPDH (bottom). NS not significant, $P = 0.4812$, two-sided $t$-test, $n = 4$ for each group, data were normalized to controls. Data are presented as mean values ± SEM. **e** Schematic diagram showing the dendrites of Drd2-positive neurons filled with biocytin in layer 5 of ACC. We analyzed the spines of the secondary basal dendrites. **f** Representative dendritic spines of Drd2-positive neurons from different aged control and SR-Drd2$^{+/-}$ rats. Scale bar, 5 μm. **g** Deficient synaptic pruning but not synapse formation of Drd2-positive neurons in SR-Drd2$^{+/-}$ rats. The spine densities of Drd2-positive neurons in different aged control and SR-Drd2$^{+/-}$ rats were quantified. NS not significant, P (2w) = 0.9718, two-way ANOVA followed by Sidak's multiple comparisons test, $n = 32$ dendrites from 5 control rats, $n = 37$ dendrites from 5 SR-Drd2$^{+/-}$ rats. P (3w) = 0.9831, two-way ANOVA followed by Sidak's multiple comparisons test, $n = 19$ dendrites from 4 control rats, $n = 26$ dendrites from 5 SR-Drd2$^{+/-}$ rats. P (4w) = 0.6744, two-way ANOVA followed by Sidak's multiple comparisons test, $n = 13$ dendrites from 3 control rats, $n = 22$ dendrites from 4 SR-Drd2$^{+/-}$ rats. *P = 0.0133, two-way ANOVA followed by Sidak's multiple comparisons test, $n = 12$ dendrites from 3 control rats, $n = 23$ dendrites from 4 SR-Drd2$^{+/-}$ rats. ***P < 0.0001, two-way ANOVA followed by Sidak's multiple comparisons test, $n = 28$ dendrites from 5 control rats, $n = 24$ dendrites from 4 SR-Drd2$^{+/-}$ rats. Data are presented as mean values ± SEM. **h** Line chart of the data in panel **g**. The statistical results are same as panel **g**. *P = 0.0133, two-way ANOVA followed by Sidak's multiple comparisons test, $n = 12$ dendrites from 3 control rats, $n = 23$ dendrites from 4 SR-Drd2$^{+/-}$ rats. ***P < 0.0001, two-way ANOVA followed by Sidak's multiple comparisons test, $n = 28$ dendrites from 5 control rats, $n = 24$ dendrites from 4 SR-Drd2$^{+/-}$ rats. Data are presented as mean values ± SEM. **i** Increased densities of mushroom-like and stubby spines in Drd2-positive neurons from SR-Drd2$^{+/-}$ rats, compared with control rats. The densities of different types of dendritic spines in Drd2-positive neurons from 8-week-old control and SR-Drd2$^{+/-}$ rats were quantified. NS not significant, $P = 0.5398$, **P = 0.0068, ***P = 0.0008, two-way ANOVA followed by Sidak's multiple comparisons test, $n = 28$ dendrites from 5 control rats, $n = 24$ dendrites from 4 SR-Drd2$^{+/-}$ rats. Data are presented as mean values ± SEM.

lead to an increase of dendritic spine density in adult SR-Drd2$^{+/-}$ rats, compared with control rats (Fig. 2f–h). Further analysis of the spine morphology indicated that the densities of mushroom-like and stubby spines in Drd2-positive neurons were increased in adult SR-Drd2$^{+/-}$ rats, compared with control rats (Fig. 2i and Supplementary Fig. 4). The higher densities of mushroom-like spines suggest that mature synapses might be increased in Drd2-positive neurons from SR-Drd2$^{+/-}$ rats, compared with controls. The increase of stubby spines might suggest that Drd2 is also involved with spinogenesis.

**Hyperglutamatergic function in the ACC of SR-Drd2$^{+/-}$ rats.** To study whether the increased dendritic spines in adult SR-Drd2$^{+/-}$ rats are functional, we performed whole-cell patch-clamp recordings of Drd2-positive neurons at different time-points (Fig. 3a). Both the frequency and amplitude of miniature excitatory postsynaptic currents (mEPSC) of Drd2-positive neurons are similar between 4-week-old SR-Drd2$^{+/-}$ and control rats (Fig. 3b–f), which is consistent with the similar dendritic spines of Drd2-positive neurons between the two genotypes (Fig. 2f–h). However, the frequency but not the amplitude of mEPSC in Drd2-positive neurons became higher in SR-Drd2$^{+/-}$ rats starting from 5-week-old age, compared with control rats (Fig. 3g–k). The increased mEPSC frequency of Drd2-positive neurons persisted in the adult SR-Drd2$^{+/-}$ rats (Fig. 3l–p), which suggested that the increased dendritic spines are functional. By contrast, both of the frequency and amplitude of miniature inhibitory postsynaptic currents (mIPSC) of Drd2-positive neurons were not changed in adult SR-Drd2$^{+/-}$ rats, compared to control rats (Supplementary Fig. 5), which indicated that Drd2$^{+/-}$ mutation may not affect the number of inhibitory synapses in Drd2-positive neurons.

In addition to differences in dendritic spine densities, we tested whether reduced Drd2 expression affected cell-intrinsic excitability as revealed by responses to direct depolarizing current injections. To this end, we performed whole-cell patch-clamp recordings of Drd2-positive neurons. As shown in Fig. 3q, r, the Drd2-positive neurons showed an upright shift of input–output (I/O) curves of action potential (AP) in SR-Drd2$^{+/-}$ rats, compared with control rats, suggesting an enhanced excitability of Drd2-positive neurons in SR-Drd2$^{+/-}$ rats. All these results demonstrate that downregulation of Drd2 impairs synaptic pruning and increases the cell-intrinsic excitability of Drd2-positive neurons.

**Activation of mTOR signaling by Drd2$^{+/-}$ in synaptic pruning.** In the following study, we address the molecular mechanisms by which DRD2 regulates synaptic pruning. The downstream of DRD2 includes cAMP-PKA and β-arrestin-PP2A-AKT signaling[30]. Downregulation of DRD2 has been shown to activate AKT-mTOR (mammalian target of rapamycin) signaling (Fig. 4a)[41,42], and activation of mTOR signaling has been implicated in the deficient synaptic pruning in the autism-spectrum diseases[9]. Due to these facts, we hypothesized activation of mTOR signaling was also involved in the impaired synaptic pruning in SR-Drd2$^{+/-}$ rats. To test this hypothesis, we first determined whether AKT-mTOR signaling was activated in the deep layers of ACC from 3- to 4-week-old SR-Drd2$^{+/-}$ rats by western blot analysis. The protein levels of p-AKT, p-mTOR, and p-S6 ribosomal protein were significantly increased in the ACC from SR-Drd2$^{+/-}$ rats, compared with controls (Fig. 4b–e). By contrast, the protein levels of total AKT, mTOR, and S6 ribosomal protein were not altered in the ACC from SR-Drd2$^{+/-}$ rats, compared to controls (Fig. 4b, f–h). These results indicate that AKT-mTOR signaling is activated in the ACC of SR-Drd2$^{+/-}$ rats during 3–4-week-old, a period just prior to synaptic elimination.

To study whether the activation of mTOR was involved in the impaired synaptic pruning from the SR-Drd2$^{+/-}$ rats, we performed daily microinjection of rapamycin (Rap, the mTOR inhibitor, 1 μM in 0.5 μL per side[43]) or vehicle into deep layers of ACC from control and SR-Drd2$^{+/-}$ rats, and the spines and mEPSC of Drd2-positive neurons were analyzed at 8-week-old age (Fig. 4i–k). While rapamycin did not affect the spine densities in Drd2-positive neurons from control rats, rapamycin significantly reduced the spine densities in Drd2-positive neurons from the SR-Drd2$^{+/-}$ rats to control levels (Fig. 4l–n). Accordingly, the increased mEPSC frequency of Drd2-positive neurons from the SR-Drd2$^{+/-}$ rats also returned to control levels after treatment with rapamycin (Fig. 4o–q). By contrast, rapamycin did not influence the mEPSC amplitude of Drd2-positive neurons from control and SR-Drd2$^{+/-}$ rats (Fig. 4o, r–s). Together, these results provide evidence that activation of mTOR signaling is involved in the impaired synaptic pruning from the SR-Drd2$^{+/-}$ rats.

**Cell-autonomous roles of Drd2 in synaptic pruning.** Drd2$^{+/-}$ mutation should have reduction in DRD2 expression globally. To demonstrate whether Drd2 regulates synaptic pruning through cell-autonomous mechanisms, we injected adeno-associated virus (AAV) expressing Cre-dependent control or Drd2 shRNA (short hairpin RNA) and EGFP into the deep layers of ACC from 3-week-old Drd2-Cre rats, and the spines and mEPSC of Drd2-positive neurons were analyzed at 8-week-old age (Fig. 5a). This strategy can downregulate Drd2 expression and label Drd2-positive cells in the ACC but not globally, which is named self-reporting Drd2 knockdown (Drd2-KD) (Fig. 5b). The protein levels of DRD2 but not DRD1 were reduced at 5-week-old age in the ACC of Drd2-Cre rats treated with AAV expression Cre-dependent Drd2 shRNA, compared with Drd2-Cre rats treated with AAV expressing Cre-dependent control shRNA (i.e., control rats) (Fig. 5c, d). The spine densities including that of mushroom-like spines were significantly increased in Drd2-positive neurons from the Drd2-KD rats, compared with control rats (Fig. 5e–g). In accord, the mEPSC frequency but not amplitude was significantly increased in Drd2-positive neurons from the Drd2-KD rats, compared with control rats (Fig. 5i–l). In agreement, the cell-intrinsic excitability was also elevated in Drd2-positive neurons from the Drd2-KD rats, compared with control rats (Fig. 5m, n). Taken together, these results indicate that Drd2 regulates synaptic pruning through a cell-autonomous manner.

**Critical roles of Drd2 in LTD of NMDAR transmission.** Synaptic pruning is thought to include some similar cellular/biochemical processes as LTD[32]. Pyramidal neurons in layer 5 of ACC receive input from layers 2–3 as well as the thalamus[44–46]. We performed theta frequency stimulation (TFS) in layers 2–3 and recorded LTD in layer 5 pyramidal neurons of ACC between 4 and 5 weeks old (Fig. 6a), the sensitive period of dendritic spine pruning (Fig. 2f–h). TFS could not induce LTD of AMPA receptor (AMPAR)-mediated transmission in layer 5 pyramidal neurons of ACC (Supplementary Fig. 6), consistent with a previous report[47]. However, TFS resulted in LTD of NMDA receptor (NMDAR)-mediated transmission in layer 5 pyramidal neurons of ACC (Fig. 6b, c). Remarkably, the LTD of NMDAR is severely impaired in Drd2-positive neurons from SR-Drd2$^{+/-}$ rats, compared with controls (Fig. 6b). These results suggest that Drd2 is important for LTD of NMDAR in Drd2-positive neurons, which is consistent with a previous report[48]. By contrast, Drd2$^{+/-}$ mutation did not compromise LTD of NMDAR in Drd2-negative neurons (Fig. 6c). To address whether LTD of NMDAR is acutely regulated by DRD2, we treated the ACC brain slices from 4-week-old SR-Drd2 rats with vehicle or 10 μM Eticlopride[49] (Eti, a DRD2 antagonist) for 20 min

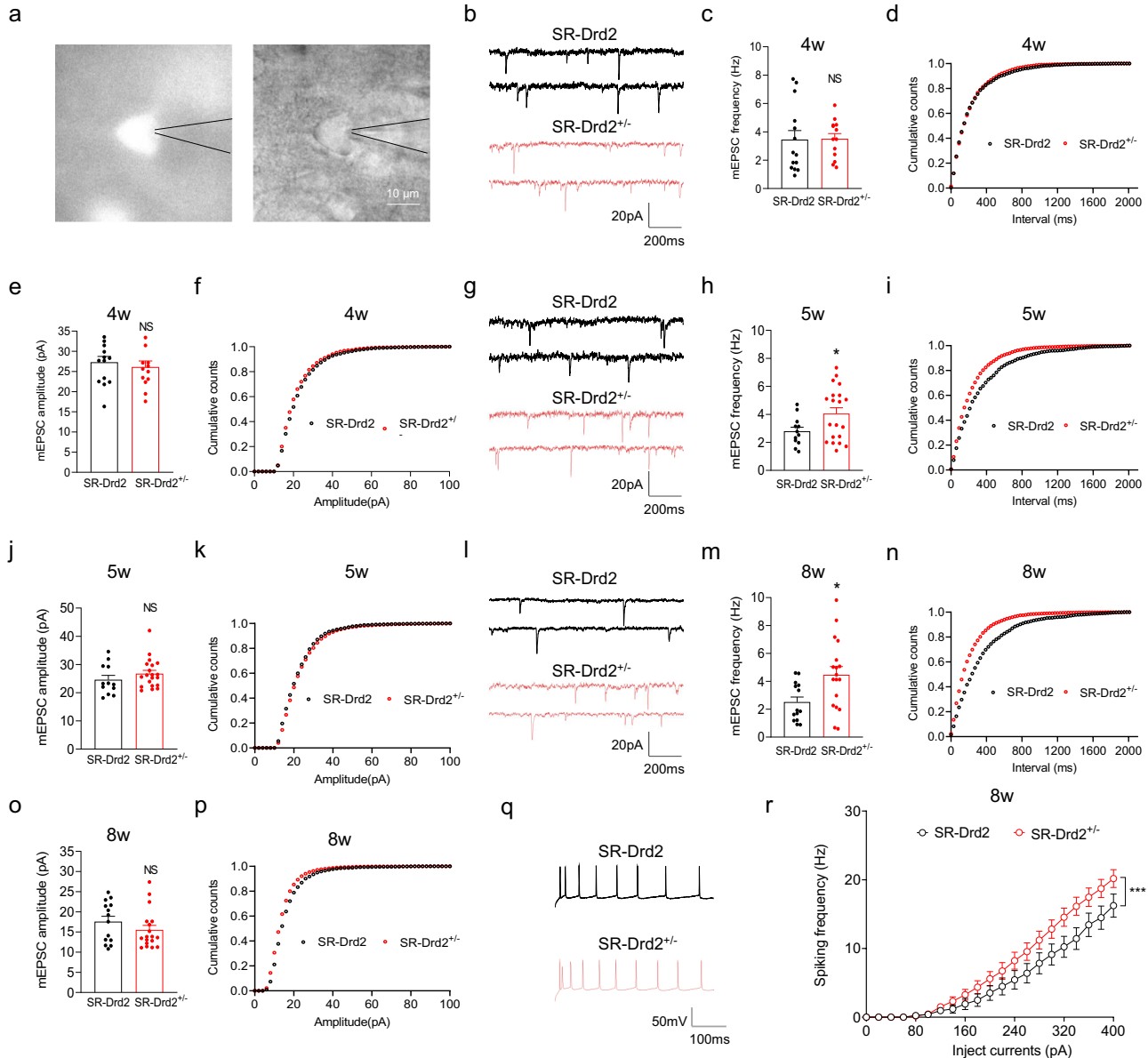

**Fig. 3 Hyperglutamatergic function of Drd2-positive neurons in SR-Drd2$^{+/-}$ rats. a** Diagram to show the whole-cell recording of Drd2-positive neurons in layer 5 of ACC from SR-Drd2 rats. Left, the fluorescent image, right, the phase-contrast image. Scale bar, 10 μm. Three independent experiments were repeated to get similar results. **b** Representative mEPSC traces of Drd2-positive neurons from 4-week-old control and SR-Drd2$^{+/-}$ rats. **c** Similar mEPSC frequency of Drd2-positive neurons between 4-week-old control and SR-Drd2$^{+/-}$ rats. NS not significant, $P = 0.9421$, two-sided $t$-test, $n = 14$ neurons from 3 control rats, $n = 13$ neurons from 3 SR-Drd2$^{+/-}$ rats. Data are presented as mean values ± SEM. **d** Cumulative plots of mEPSC frequency. **e** Similar mEPSC amplitude of Drd2-positive neurons between 4-week-old control and SR-Drd2$^{+/-}$ rats. NS not significant, $P = 0.5763$, two-sided $t$-test, $n = 14$ neurons from 3 control rats, $n = 13$ neurons from 3 SR-Drd2$^{+/-}$ rats. Data are presented as mean values ± SEM. **f** Cumulative plots of mEPSC amplitude. **g** Representative mEPSC traces of Drd2-positive neurons from 5-week-old control and SR-Drd2$^{+/-}$ rats. **h** Increased mEPSC frequency of Drd2-positive neurons in 5-week-old SR-Drd2$^{+/-}$ rats. *$P = 0.0435$, two-sided $t$-test, $n = 12$ neurons from 3 control rats, $n = 20$ neurons from 4 SR-Drd2$^{+/-}$ rats. Data are presented as mean values ± SEM. **i** Cumulative plots of mEPSC frequency. **j** Similar mEPSC amplitude of Drd2-positive neurons between 5-week-old control and SR-Drd2$^{+/-}$ rats. NS not significant, $P = 0.263$, two-sided $t$-test, $n = 12$ neurons from 3 control rats, $n = 20$ neurons from 4 SR-Drd2$^{+/-}$ rats. Data are presented as mean values ± SEM. **k** Cumulative plots of mEPSC amplitude. **l** Representative mEPSC traces of Drd2-positive neurons from 8-week-old control and SR-Drd2$^{+/-}$ rats. **m** Increased mEPSC frequency of Drd2-positive neurons in 8-week-old SR-Drd2$^{+/-}$ rats. *$P = 0.0139$, two-sided $t$-test, $n = 14$ neurons from 3 control rats, $n = 18$ neurons from 4 SR-Drd2$^{+/-}$ rats. Data are presented as mean values ± SEM. **n** Cumulative plots of mEPSC frequency. **o** Similar mEPSC amplitude of Drd2-positive neurons between 8-week-old control and SR-Drd2$^{+/-}$ rats. NS not significant, $P = 0.2503$, two-sided $t$-test, $n = 14$ neurons from 3 control rats, $n = 18$ neurons from 4 SR-Drd2$^{+/-}$ rats. Data are presented as mean values ± SEM. **p** Cumulative plots of mEPSC amplitude. **q** Representative action potentials of Drd2-positive neurons from 8-week-old control and SR-Drd2$^{+/-}$ rats. **r** Increased excitability of Drd2-positive neurons in 8-week-old SR-Drd2$^{+/-}$ rats, compared with control rats. Shown are the I/O curves of action potentials from Drd2-positive neurons. *** Genotype F (1, 2010) = 57.55, $P < 0.0001$, two-way ANOVA, $n = 25$ neurons from 5 control rats, $n = 44$ neurons from 7 SR-Drd2$^{+/-}$ rats. Data are presented as mean values ± SEM.

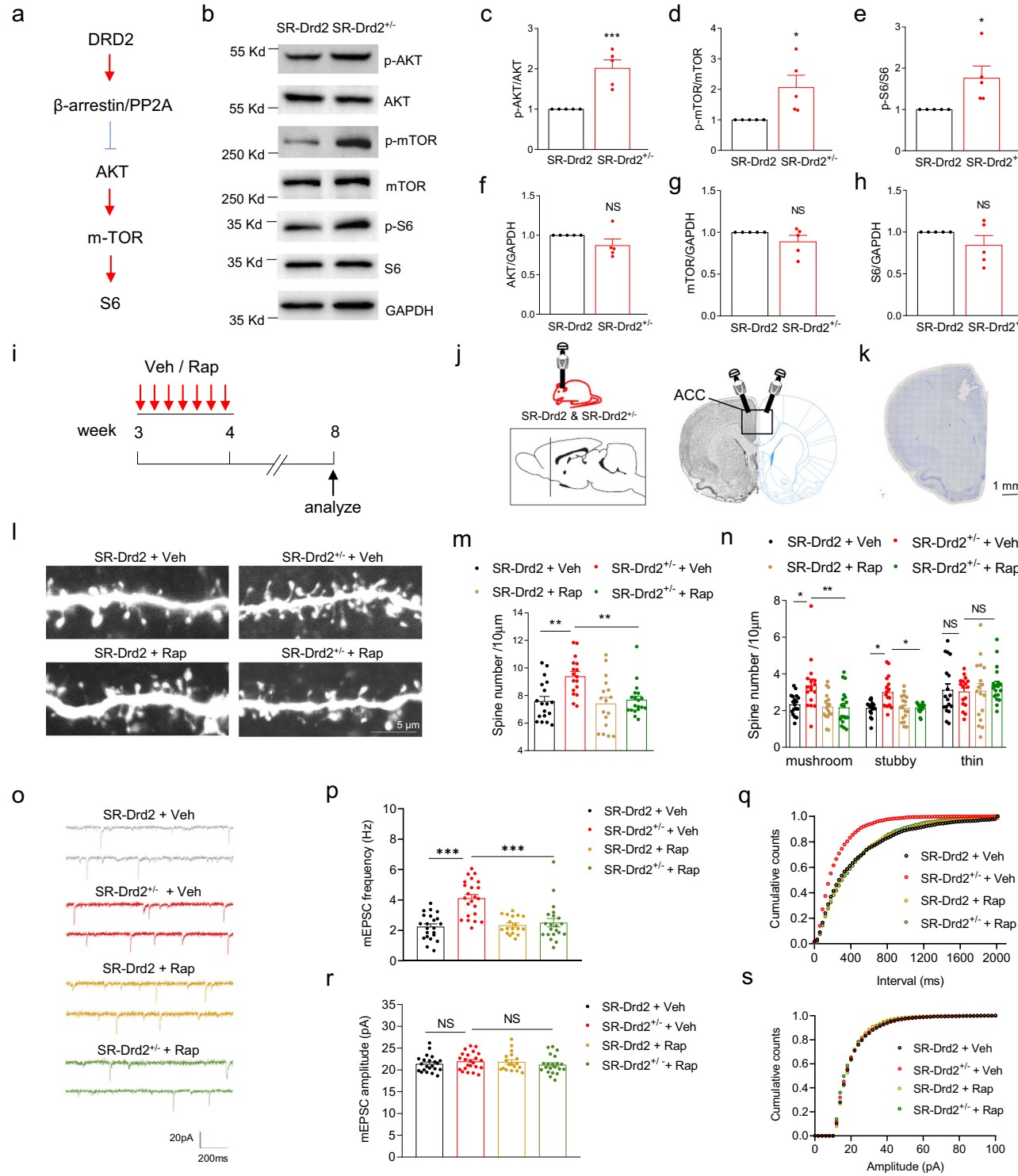

and then recorded LTD on pyramidal neurons in layer 5 of ACC. Intriguingly, acute inhibition of DRD2 impaired LTD of NMDAR in Drd2-positve but not Drd2-negative neurons (Fig. 6d, e).

To study whether Drd2 heterozygous mutation affected dendritic spines in Drd2-negative neurons, we filled the Drd2-negative neurons with biocytin and used immunofluorescence to reveal the dendritic spines (Fig. 6f, g). We analyzed the spines from the secondary basal dendrites of Drd2-negative neurons as we did for Drd2-positive neurons. As contrary to the increased spine density in Drd2-positive neurons (Fig. 2f–i), the spine

density of Drd2-negative neurons was not altered in adult SR-Drd2[+/−] rats, compared to control rats (Fig. 6h and i). Altogether, these results indicate that Drd2 regulates LTD and dendritic spine pruning only in Drd2-positive neurons but not in Drd2-negative neurons.

**Sensitive periods in Drd2 regulation of synaptic pruning.** In the following study, we investigate whether Drd2 regulates synaptic pruning through sensitive periods. Toward this aim, we

**Fig. 4 Activation of mTOR signaling by Drd2$^{+/-}$ in synaptic pruning. a** DRD2 recruits β-arrestin/PP2A complex to inhibit AKT-mTOR-S6 signaling pathway. **b** Increased AKT-mTOR-S6 signaling in the deep layers of ACC from 3–4-week-old SR-Drd2$^{+/-}$ rats, compared with control rats. The homogenates from deep layers of ACC were subjected to western blot and probed with the indicated antibodies. **c** Quantification of p-AKT/AKT in panel **b**. ***$P = 0.0009$, two-sided $t$-test, $n = 5$ for each group, data were normalized to controls. **d** Quantification of p-mTOR/mTOR in panel **b**. *$P = 0.0249$, two-sided $t$-test, $n = 5$ for each group, data were normalized to controls. **e** Quantification of p-S6/S6 in panel **b**. *$P = 0.0291$, two-sided $t$-test, $n = 5$ for each group, data were normalized to controls. **f** Quantification of AKT/GAPDH in panel **b**. NS not significant, $P = 0.1459$, two-sided $t$-test, $n = 5$ for each group, data were normalized to controls. **g** Quantification of mTOR/GAPDH in panel **b**. NS not significant, $P = 0.1793$, two-sided $t$-test, $n = 5$ for each group, data were normalized to controls. **h** Quantification of S6/GAPDH in panel **b**. NS not significant, $P = 0.2099$, two-sided $t$-test, $n = 5$ for each group, data were normalized to controls. Data are presented as mean values ± SEM. **i** Experimental design. Control and SR-Drd2$^{+/-}$ rats received daily injection of rapamycin (1 μM in 0.5 μl per side) or Veh (0.5 μl saline per side) into layer 5 of ACC between 3- and 4-week-old age, and the spine densities and mEPSC of Drd2-positive neurons were analyzed at 8-week-old age. **j** Diagram showing the injection of Veh or Rap into layer 5 of ACC in control and SR-Drd2$^{+/-}$ rat. The dashed line of the sagittal section diagram indicates the position of the coronal section. The rectangle indicates the brain regions of ACC. **k** Nissl staining to determine the injection sites. Scale bar, 1 mm. **l** Rapamycin rescued the increased spine densities of Drd2-positive neurons from SR-Drd2$^{+/-}$ rat. Shown were the representative spines of Drd2-positive neurons from different groups of mice. **m** Quantification of spine densities from Drd2-positive neurons in panel **l**. **$P$ (SR-Drd2 + Veh vs SR-Drd2$^{+/-}$ + Veh) = 0.0013, **$P$ (SR-Drd2$^{+/-}$ + Veh vs SR-Drd2$^{+/-}$ + Rap) = 0.0019, mixed-effects analysis followed by Sidak's multiple comparisons test. $n = 18$ dendrites from 3 SR-Drd2 rats treated with Veh, $n = 17$ dendrites from 3 SR-Drd2$^{+/-}$ rats treated with Veh, $n = 17$ dendrites from 3 SR-Drd2 rats treated with Rap, $n = 19$ dendrites from 3 SR-Drd2$^{+/-}$ rats treated with Rap. Data are presented as mean values ± SEM. **n** Quantification of densities of different types of spines from Drd2-positive neurons in panel **l**. *$P$ (mushroom) = 0.0211, **$P$ (mushroom) = 0.0041, *$P$ (stubby, SR-Drd2 + Veh vs SR-Drd2$^{+/-}$ + Veh) = 0.0421, *$P$ (stubby, SR-Drd2$^{+/-}$ + Veh vs SR-Drd2$^{+/-}$ + Rap) = 0.0435, two-way ANOVA followed by Turkey's multiple comparisons test. $n = 18$ dendrites from 3 SR-Drd2 rats treated with Veh, $n = 17$ dendrites from 3 SR-Drd2$^{+/-}$ rats treated with Veh, $n = 17$ dendrites from 3 SR-Drd2 rats treated with Rap, $n = 19$ dendrites from 3 SR-Drd2$^{+/-}$ rats treated with Rap. Data are presented as mean values ± SEM. **o** Rapamycin rescued the increased mEPSC frequency of Drd2-positive neurons from SR-Drd2$^{+/-}$ rat. Shown were the representative mEPSC traces of Drd2-positive neurons from different groups of rats. **p** Quantification of mEPSC frequency of Drd2-positive neurons in panel **o**. ***$P < 0.0001$, two-way ANOVA followed by Turkey's multiple comparisons test. $n = 22$ neurons from 4 SR-Drd2 rats treated with Veh, $n = 23$ neurons from 4 SR-Drd2$^{+/-}$ rats treated with Veh, $n = 18$ neurons from 4 SR-Drd2 rats treated with Rap, $n = 21$ neurons from 4 SR-Drd2$^{+/-}$ rats treated with Rap. Data are presented as mean values ± SEM. **q** Cumulative plots of mEPSC frequency. **r** Rapamycin did not affect mEPSC amplitude of Drd2-positive neurons in control and SR-Drd2$^{+/-}$ rats. Shown were the quantification of mEPSC frequency of Drd2-positive neurons in panel **o**. NS not significant, $P = 0.8442$, SR-Drd2 + Veh vs SR-Drd2$^{+/-}$ + Veh, $P = 0.6968$, SR-Drd2$^{+/-}$ + Veh vs SR-Drd2$^{+/-}$ + Rap, two-way ANOVA followed by Turkey's multiple comparisons test. $n = 22$ neurons from 4 SR-Drd2 rats treated with Veh, $n = 23$ neurons from 4 SR-Drd2$^{+/-}$ rats treated with Veh, $n = 18$ neurons from 4 SR-Drd2 rats treated with Rap, $n = 21$ neurons from 4 SR-Drd2$^{+/-}$ rats treated with Rap. Data are presented as mean values ± SEM. **s** Cumulative plots of mEPSC amplitude.

performed daily microinjection of Eti (1 μg in 0.5 μL per side)[50], a Drd2 antagonist or vehicle (Veh) into deep layers of ACC in SR-Drd2 rats between 3 and 4 weeks old (Fig. 7a–c), the period just prior to the synaptic elimination (Fig. 2f–h). The spine density of Drd2-positive neurons was similar between 4-week-old SR-Drd2 rats treated with Veh or Eti (Fig. 7d–f), which suggest that DRD2 inhibition may not affect synapse formation. However, the spine density of Drd2-positive neurons was significantly higher in 5- and 8-week-old SR-Drd2 rats treated with Eti, compared with controls (Fig. 7d–f). Further analysis of the spine morphology indicated an increase of mushroom-like dendritic spines in Drd2-positive neurons after Eti treatment, compared with controls, in adulthood (Fig. 7g). By contrast, daily injection of Eti into deep layers of ACC between 7 and 8 weeks old did not alter the dendritic spine density in Drd2-positive neurons (Supplementary Fig. 7a–f). Taken together, these results demonstrate that Drd2 regulates synaptic pruning in sensitive periods during adolescence.

**Hyperglutamatergic function after inhibition of Drd2 during adolescence.** Next, we illustrate whether the deficits in Drd2-mediated synaptic pruning altered glutamatergic transmission in adulthood. To this end, we performed daily injection of Veh or Eti into layer 5 of ACC in SR-Drd2 rats between 3 and 4 weeks old, and the spontaneous excitatory postsynaptic currents (sEPSC) of Drd2-positive neurons were analyzed at 8 weeks old (Fig. 8a). Application of Eti between 3 and 4 weeks old increased the frequency but not the amplitude of sEPSC in Drd2-positive neurons, compared with controls (Fig. 8b–f). By contrast, treatment with Eti between 3 and 4 weeks old did not alter the frequency or amplitude of spontaneous inhibitory postsynaptic currents (sIPSC) in Drd2-positive neurons (Fig. 8g–k). These results suggest that inhibition of DRD2 during 3–4-weeks-old age

would increase glutamatergic transmission on Drd2-positive neurons in 8-weeks-old age. Remarkably, inhibition of DRD2 during 3–4-weeks-old age also elevated the cell excitability of Drd2-positive neurons in 8-weeks-old age (Fig. 8l and m). By contrast, treatment with Eti between 7 and 8 weeks old did not change the frequency or amplitude of sEPSC in Drd2-positive neurons (Supplementary Fig. 7g–k), which is in line with the finding that administration of Eti between 7 and 8 weeks old did not alter the spine density of Drd2-positive neurons (Supplementary Fig. 7a–f). Taken together, these results demonstrate that inhibition of DRD2 during adolescence led to hyperglutamatergic function of Drd2-positive neurons in adulthood.

**Anxiety-like behaviors after inhibition of DRD2 in the ACC during adolescence.** Lastly, we investigate whether inhibition of DRD2 in the ACC during adolescence causes behavioral deficits in adulthood. Toward this goal, we performed daily injection of Veh or Eti into layer 5 of ACC in WT rats between 3 and 4 weeks old and analyzed the behaviors in 8 weeks old (Fig. 9a). Inhibition of DRD2 in layer 5 of ACC during 3–4 weeks old did not affect locomotion in the open field for adult rats (Fig. 9b, c). ACC is implicated in regulating anxiety and social behavior[36,51,52]. We next performed experiments in light-dark box and elevated O-maze to evaluate the anxiety-like behaviors[53]. Intriguingly, the rats spent lesser time in the light box and open arm after inhibition of DRD2 in layer 5 of ACC during 3–4 weeks old (Fig. 9d–g), indicating elevated anxiety-like behaviors. By contrast, we did not observe social behavioral deficits in the three-chamber tests after inhibition of DRD2 in layer 5 of ACC during 3–4 weeks old (Fig. 9h–k). Of note, inhibition of DRD2 in layer 5 of ACC between 7- and 8-week-old age did not affect locomotion, anxiety-like, or social behaviors (Supplementary Fig. 8). Taken together, these results demonstrate that inhibition of DRD2 in

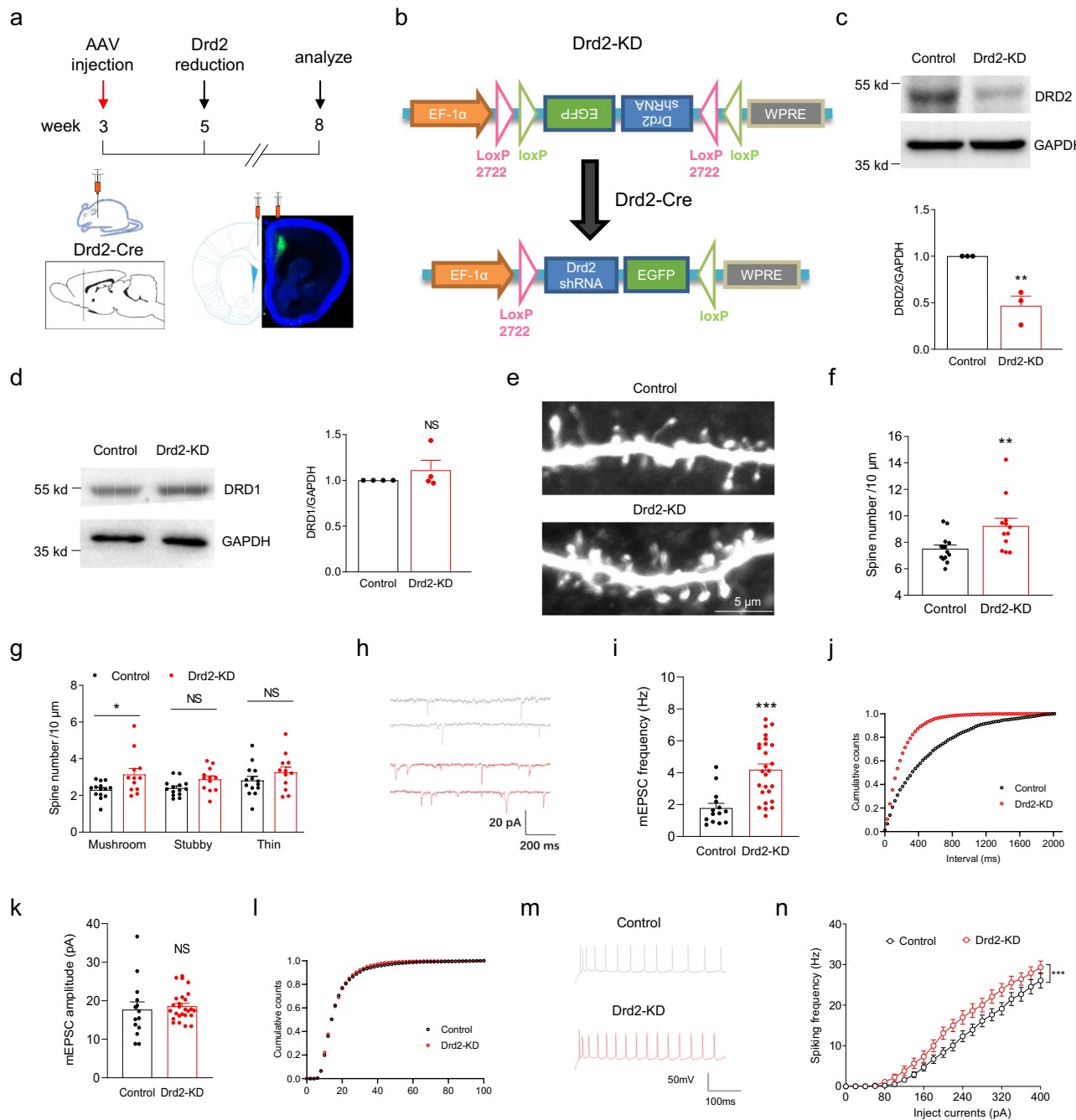

layer 5 of ACC during adolescence leads to elevated anxiety-like behaviors in adulthood.

## Discussion

In this study, we demonstrate important roles of DRD2 in cortical synaptic pruning through time course studies from the control and SR-Drd2$^{+/-}$ rats. We also show that DRD2 is important for LTD, a form of synaptic plasticity which includes some similar cellular/biochemical processes as synaptic pruning. We further demonstrate that DRD2 regulates synaptic pruning via cell-autonomous mechanisms including activation of mTOR signaling in sensitive periods. We lastly illustrate that inhibition of DRD2 in the ACC during adolescence leads to hyperglutamatergic function and anxiety-like behaviors in adulthood. These results demonstrate mechanisms underlying synaptic pruning and

how the deficits of these processes cause brain dysfunction in adulthood.

A previous study found reduced spine density in the pyramidal neurons from PFC of Drd2 null mutant mice[54]. However, the Drd2 null mutant mice suffer from growth retardation, which complicates the interpretation of the results. Since the SR-Drd2$^{+/-}$ rats show normal body and brain weight (Supplementary Fig. 2a–d), the deficient synaptic pruning in SR-Drd2$^{+/-}$ rats may not be secondary to the growth retardation. Another previous study showed that knockdown of Drd2 increased dendritic spine density in hippocampal pyramidal neurons[55], which is consistent with our findings in the ACC. However, the previous works of Drd2 in dendritic spine development did not perform time course studies or differentiate between Drd2-positive and Drd2-negative neurons. The latter issue is especially important considering the heterogeneity of cortical and hippocampal

**Fig. 5 Cell-autonomous roles of Drd2 in synaptic pruning. a** Experimental design. 3-week-old Drd2-Cre rats received daily injection of AAV expressing Cre-dependent control or Drd2 shRNA and EGFP into deep layers of ACC, and the spine densities, mEPSC, and excitability of Drd2-positive neurons were analyzed at 8-week-old age. **b** The diagram showing Cre-dependent expression of Drd2 shRNA and EGFP. This strategy could simultaneously knockdown Drd2 expression and label Drd2-positive neurons, and was named self-reporting Drd2 knockdown (Drd2-KD). **c** Decreased DRD2 protein levels in the deep layers of ACC from SR-Drd2-KD rats. The homogenates of deep layers of ACC from control and Drd2-KD rats were subjected to western blot and probed with anti-DRD2 and anti-GAPDH antibodies (top). Quantification of DRD2/GAPDH (bottom). **$P = 0.007$, two-sided $t$-test, $n = 3$ for each group, data were normalized to controls. Data are presented as mean values ± SEM. **d** Unchanged DRD1 protein levels in the deep layers of ACC from Drd2-KD rats. The homogenates of deep layers of ACC from control and Drd2-KD rats were subjected to western blot and probed with anti-DRD1 and anti-GAPDH antibodies (left). Quantification of DRD1/GAPDH (right). NS, not significant, $P = 0.3531$, two-sided $t$-test, $n = 4$ for each group, data were normalized to controls. Data are presented as mean values ± SEM. **e** Increased spine densities of Drd2-positive neurons in layer 5 of ACC from Drd2-KD rats, compared with control rats. Shown were the representative spines of Drd2-positive neurons. **f** Quantification of spine densities from Drd2-positive neurons in panel **e**. **$P = 0.01$, two-tailed $t$-test, $n = 14$ dendrites from 3 control rats, $n = 12$ dendrites from 3 Drd2-KD rats. Data are presented as mean values ± SEM. **g** Quantification of densities of different types of spines from Drd2-positive neurons in panel **e**. NS not significant, *$P = 0.0225$, two-way ANOVA followed by Sidak's multiple comparisons test. $n = 14$ dendrites from 3 control rats, $n = 12$ dendrites from 3 Drd2-KD rats. Data are presented as mean values ± SEM. **h** Representative mEPSC traces of Drd2-positive neurons from 8-week-old control and Drd2-KD rats. **i** Increased mEPSC frequency of Drd2-positive neurons in 8-week-old Drd2-KD rats, compared with control rats. ***$P < 0.0001$, two-sided $t$-test, $n = 14$ neurons from 3 control rats, $n = 26$ neurons from 5 SR-Drd2$^{+/-}$ rats. Data are presented as mean values ± SEM. **j** Cumulative plots of mEPSC frequency. **k** Similar mEPSC amplitude of Drd2-positive neurons between 8-week-old control and Drd2-KD rats. NS not significant, $P = 0.6173$, two-sided $t$-test, $n = 14$ neurons from 3 control rats, $n = 26$ neurons from 5 Drd2-KD rats. Data are presented as mean values ± SEM. **l** Cumulative plots of mEPSC amplitude. **m** Representative action potentials of Drd2-positive neurons from 8-week-old control and Drd2-KD rats. **n** Increased excitability of Drd2-positive neurons in 8-week-old Drd2-KD rats, compared with control rats. Shown are the I/O curves of action potentials from Drd2-positive neurons. *** Genotype $F_{(1, 945)} = 52.67$, $P < 0.0001$, two-way ANOVA, $n = 29$ neurons from 5 control rats, $n = 18$ neurons from 4 Drd2-KD rats. Data are presented as mean values ± SEM.

neurons. For instance, in the hippocampus, Drd2 is mainly expressed in GABAergic interneurons in the CA1-CA3 regions while it is highly distributed in excitatory granule neurons in the dentate gyrus[38,56].

DRD2 is expressed at dopaminergic terminals to regulate dopamine release[57,58]. DRD2 is also present on dendrites of pyramidal neurons, where it suppresses long-term potentiation (LTP)[59]. Here we show that DRD2 is important for LTD of NMDAR transmission in layer 5 Drd2-positive neurons in ACC (Fig. 6b). Intriguingly, LTD signaling pathway has been shown to be required for NMDAR-dependent synaptic pruning[60]. By contrast, both LTD of NMDAR transmission and spine densities are normal in Drd2-negative neurons from SR-Drd2$^{+/-}$ rats (Fig. 6c and 6f–i). It has been proposed that LTD onto hippocampal CA1 pyramidal neurons is mediated by presynaptic DRD2[61]. However, the cellular expression pattern of Drd2 is different between hippocampus and ACC. Drd2 is not expressed postsynaptically in CA1 pyramidal neurons in the dorsal hippocampus[38,56,61], while Drd2 is highly expressed in layer 5 pyramidal neurons in the ACC (Fig. 1c and d). Taken together, these results suggest that DRD2 regulates LTD through different mechanisms in the hippocampus and ACC.

The deficits in synaptic pruning may result in excessive glutamatergic transmission and hyperactivation of pyramidal neurons in adulthood. However, the hyperexcitable Drd2-positive neurons from SR-Drd2$^{+/-}$ rats might also be due to changes in properties of hyperpolarization-activated, cyclic nucleotide-gated (HCN) channels or other voltage-gated ion channels[49,62,63]. The hyperactivity of ACC has been shown to drive anxiety-like behaviors[64,65]. In agreement, the common anti-anxiety medicines such as benzodiazepines act by enhancing the GABAergic transmission and inhibiting neural activity[66]. These results are consistent with our findings that inhibition of DRD2 in ACC during adolescence lead to hyperactivation of pyramidal neurons (Fig. 8) and elevated anxiety-like behaviors (Fig. 9). Interestingly, the Drd2 A1 allele, a Drd2 gene polymorphism, which causes lower Drd2 expression levels in the brain, has been associated with anxiety and depression[67,68]. In addition to anxiety, ACC has also been implicated in social interaction[51]. However, the recent study indicated that hypofunction but not hyperfunction of pyramidal neurons in the ACC may cause social deficits[69]. Here we did not analysis behaviors in Drd2$^{+/-}$ rats because the global

reduction of Drd2 expression in Drd2$^{+/-}$ rats made the behavioral results difficult to be interpreted. However, the behavioral effects could be due to changes in presynaptic dopamine release through presynaptic D2R projecting into the ACC, independently of the synaptic pruning.

To date, most of the studies on molecular mechanisms of synapse development were from mice because of the availability of the vast genetic toolbox for mice. However, with larger brains and ability to perform more complex behavioral paradigms, rats can provide a better translational validity for brain disorders than mice[70]. Here we used genetic mutant rats to demonstrate an important role of Drd2 in synaptic pruning in basal dendrites of pyramidal neurons in deep layers of ACC. Intriguingly, schizophrenia is thought to be due to the excessive synaptic pruning in the cerebral cortex[8,10,71]. Here we show that DRD2 antagonist attenuated cortical synaptic pruning, which is hypothesized to be beneficial for schizophrenia patients. Given that Drd2 is genetically linked to schizophrenia and is the target of antipsychotics[21,23], our findings may provide insight into the pathophysiological mechanisms of schizophrenia.

## Methods

**Generation of SR-Drd2 and Drd2$^{+/-}$ rats**. SR-Drd2 rats were obtained by crossing the Drd2-Cre knockin rats and the Rosa26-LSL-tdTomato knockin rats as described in our previous study[38]. To generate Drd2$^{+/-}$ rats, the exon 3 of rat Drd2 gene was deleted using CRISPR/Cas9 technology[72]. Briefly, gRNA and Cas9 mRNA were injected into the cytoplasm of one-cell stage embryos through the injection needle. Injections were performed using an Eppendorf transferMan NK2 micromanipulator. Injected zygotes were transferred into pseudopregnant female rats after 2 h culture in KSOM medium. The primers for genotyping the Drd2$^{+/-}$ rats are as follows: forward: 5'- tcctgccattacattgatttt-3, reverse: 5'-ctctggtaa-cagtggcatca-3. The PCR products for WT and mutant Drd2 alleles are 950 and 450 bp, respectively. All the founding rat lines were on Sprague-Dawley (SD) background and were backcrossed with WT SD rats for five generations before experiments. To eliminate the possible effects of the hormone cycle, male rats were used in all experiments. For immunofluorescence, we used 8-week-old rats. For western blot to detect AKT-mTOR signaling, we used 3–4-week-old rats. For analysis of dendritic branch, we used 8-week-old rats. For analysis of dendritic spines, we used 2-, 3-, 4-, 5-, and 8-week-old rats. For electrophysiological recording, we used 4-, 5-, and 8-week-old rats. For stereotaxic injection of AAV expressing Cre-dependent shRNA, we used 3-week-old rats. For microinjection of Eti and Rap into ACC, we used 3–4-week-old and 7–8-week-old rats. For behavioral analysis, we used 8-week-old rats. Animals were housed in rooms at 23 °C and 50% humidity in a 12 h light/dark cycle and with food and water available ad libitum. Animal experimental procedures were approved by the Institutional Animal Care and Use Committee of East China Normal University.

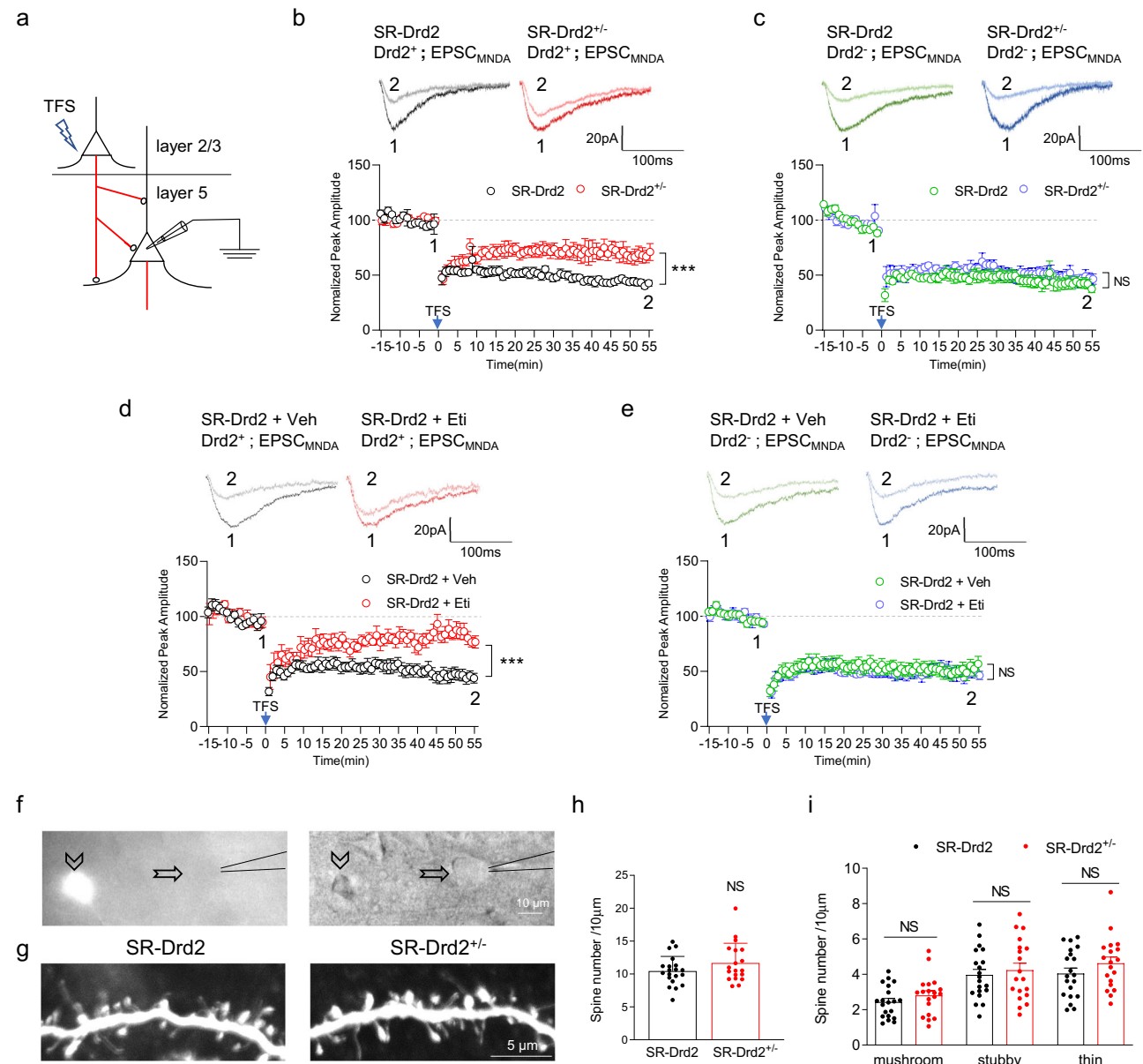

**Immunofluorescence**. The process of immunofluorescence on rat brain slices was performed as described in our previous studies[38]. Briefly, brain slices were permeabilized with 0.5% Triton-X 100 and 5% BSA in PBS and incubated with primary antibodies at 4 °C overnight.

The brain slices were not treated with Triton-X 100 when staining with anti-GAD67 antibodies. After washing with PBS for three times, samples were incubated with Alexa Fluor-488 goat-anti-mouse secondary antibodies (A11029; 1:500, Thermo Fisher) or Alexa Fluor-647 goat-anti-rabbit secondary antibodies (A21244; 1:500, Thermo Fisher) for 1 h at room temperature. Samples were mounted with Vectashield mounting medium (Vector Labs) and images were taken by Leica TCS SP8 confocal microscope. Details of the imaging: Objective: ×20 (oil); Numerical Aperture: 0.75; The pixel sizes in microns for x-y are 0.568; image size: 1024 × 1024 px; zoom: 1×; line averaging: 1; acquisition rate: 400 Hz. The following primary antibodies were used: rabbit anti-NeuN (1:500, Abcam, ab177487), and mouse anti-GAD67 (1:300, Millipore, MAB5406). Unbiased stereology TissueFAX Plus ST (Tissue Gnostics, Vienna, Austria)[38] was applied to count tdTomato-positive and NeuN-positive cells in brain slices. The detailed methods for cell number counting is available on the website (https://tissuegnostics.com/products/single-cell-analysis/tissuequest). The bounds of rat ACC were defined following the previous publication[73].

**Western blot**. Homogenates of ACC were prepared in RIPA buffer containing 50 mM Tris–HCl, pH 7.4, 150 mM NaCl, 2 mM EDTA, 1% sodium deoxycholate, 1% SDS, 1 mM PMSF, 50 mM sodium fluoride, 1 mM sodium vanadate, 1 mM DTT, and protease inhibitors cocktails. All the protein samples were boiled in 100 °C water bath for 10 min before western blot. Homogenates were resolved on

SDS/PAGE and transferred to nitrocellulose membranes, which were incubated in the TBS buffer containing 0.1% Tween-20 and 5% milk for 1 h at room temperature before the addition of primary antibody for incubation overnight at 4 °C. After wash, the membranes were incubated with HRP-conjugated secondary antibody (goat-anti-rabbit, G-21234, 1:2000, Thermo Fisher) in the same TBS buffer for 1 h at room temperature. For DRD1 detection, the membranes were preincubated for 1 h at 22–24 °C with rabbit-anti-rat IgG (1:500, Sangon, D111017) before incubation with the HRP-conjugated secondary antibody (goat-anti-rabbit, G-21234, 1:2000, Thermo Fisher). Immunoreactive bands were visualized by ChemiDocTM XRS + Imaging System (BIO-RAD) using enhanced chemiluminescence (Pierce) and analyzed with Image J (NIH). The following antibodies were used: rabbit anti-DRD2 (1:1000, Millipore, AB15588), rat anti-DRD1 (1:200, Sigma, D2944), rabbit anti-p-AKT (1:1000, arigo, ARG51558), rabbit anti-AKT (1:1000, GeneTex, GTX121937), rabbit anti-p-mTOR (1:1000, Cell Signaling Technology, 2974), rabbit anti-mTOR (1:1000, Cell Signaling Technology, 2983), rabbit anti-p-S6 ribosomal protein (Ser235/236) (1:1000, Cell Signaling Technology, 2211), rabbit anti-S6 ribosomal protein (1:1000, Cell Signaling Technology, 2217), and rabbit anti-GAPDH (1:5000, Abways, ab0037).

**Analysis of dendrites and dendritic spines**. We filled the neurons in rat brain slices with biocytin (0.5%, Sigma, B4261) through the recording pipette. The brain slices were then fixed in PBS containing 4% PFA for 24–48 h. After washing with PBS for five times, brain slices were incubated with blocking solution (PBS containing 10% goat serum and 0.5% Triton X 100) overnight at 4 °C. Then brain slices were incubated with Alexa Fluor-488 conjugated streptavidin (S32354; 1:200, Thermo Fisher) for 7 days at 4 °C. After washing with PBS for five times, brain

**Fig. 6 Critical roles of DRD2 in LTD of NMDAR transmission. a** Schematic diagram showing LTD recordings in layer 5 pyramidal neurons of ACC by application of theta frequency stimulation (TFS) in layers 2–3 of ACC. Red lines indicate axons. The rats used were 4–5-week-old age. **b** Impaired LTD of NMDAR transmission in Drd2-positive neurons from SR-Drd2$^{+/−}$ rats. Top, representative traces of NMDAR currents before TFS (1) and 55 min after TFS (2). Bottom, normalized peak amplitudes of NMDAR currents from Drd2-positive neurons were plotted every 1 min for SR-Drd2 and SR-Drd2$^{+/−}$ rats. *** Genotype F (1, 70) = 68.26, P < 0.0001, two-way ANOVA, n = 9 neurons from 5 SR-Drd2 rats, n = 7 neurons from 4 SR-Drd2$^{+/−}$ rats. Data are presented as mean values ± SEM. The magnitude of LTD was calculated at the amplitudes of NMDA currents 50–55 min after TFS. **c** Normal LTD of NMDAR transmission in Drd2-negative neurons from SR-Drd2$^{+/−}$ rats. Top, representative traces of NMDAR currents before TFS (1) and 55 min after TFS (2). Bottom, normalized peak amplitudes of NMDAR currents from Drd2-negative neurons were plotted every 1 min for SR-Drd2 and SR-Drd2$^{+/−}$ rats. NS not significant, Genotype F (1, 45) = 2.688, P = 0.1081, two-way ANOVA, n = 5 neurons from 3 SR-Drd2 rats, n = 6 neurons from 3 SR-Drd2$^{+/−}$ rats. Data are presented as mean values ± SEM. The magnitude of LTD was calculated at the amplitudes of NMDA currents 50–55 min after TFS. **d** Impaired LTD of NMDAR transmission in Drd2-positive neurons after acute inhibition of DRD2. The ACC slices from 4-week-old SR-Drd2 rats were treated with 10 μM Eti or Veh for 20 min before induction of LTD. Top, representative traces of NMDAR currents before TFS (1) and 55 min after TFS (2). Bottom, normalized peak amplitudes of NMDAR currents from Drd2-positive neurons were plotted every 1 min. *** Genotype F (1, 50) = 101.9, P < 0.0001, two-way ANOVA, n = 6 neurons from 3 SR-Drd2 rats for each group. Data are presented as mean values ± SEM. The magnitude of LTD was calculated at the amplitudes of NMDA currents 50–55 min after TFS. **e** Normal LTD of NMDAR transmission in Drd2-negative neurons after acute inhibition of DRD2. The ACC slices from 4-week-old SR-Drd2 rats were treated with 10 μM Eti or Veh for 20 min before induction of LTD. Top, representative traces of NMDAR currents before TFS (1) and 55 min after TFS (2). Bottom, normalized peak amplitudes of NMDAR currents from Drd2-negative neurons were plotted every 1 min. NS not significant, Genotype F (1, 50) = 1.901, P = 0.1741, two-way ANOVA, n = 7 neurons from 4 SR-Drd2 rats treated with Veh, n = 5 neurons from 3 SR-Drd$^−$ rats treated with Eti. Data are presented as mean values ± SEM. The magnitude of LTD was calculated at the amplitudes of NMDA currents 50–55 min after TFS. **f** Diagram to fill Drd2-negative neurons with biocytin in layer 5 of ACC from SR-Drd2 rats. Left, the fluorescent image, right, the phase-contrast image. Arrowhead indicates Drd2-positive neurons, arrow indicates Drd2-negative neurons. Scale bar, 10 μm. **g** Representative images of dendritic spines in Drd2-negative neurons from 8-week-old SR-Drd2 and SR-Drd2$^{+/−}$ rats. Scale bar, 5 μm. **h** Normal dendritic spine densities in Drd2-negative neurons from 8-week-old SR-Drd2$^{+/−}$ rats. NS, not significant, P = 0.1519, two-sided t-test, n = 20 dendrites from 3 SR-Drd2 rats, n = 19 dendrites from 3 SR-Drd2$^{+/−}$ rats. Data are presented as mean values ± SEM. **i** Normal densities of different types of dendritic spines in Drd2-negative neurons from 8-week-old SR-Drd2$^{+/−}$ rats. NS not significant, P (mushroom) = 0.7951, P (stubby) = 0.9954, P (thin) = 0.8163, two-way-ANOVA followed by Sidak's multiple comparisons test, n = 20 dendrites from 3 SR-Drd2 rats, n = 19 dendrites from 3 SR-Drd2$^{+/−}$ rats. Data are presented as mean values ± SEM.

slices were mounted with Vectashield mounting medium (Vector Labs) and images were taken by Leica TCS SP8 confocal microscope. We used ×20 oil immersion objective to capture images of pyramidal neurons, for each neuron, z-stacks were captured with the following parameters: z-step: 1 μm; image size: 1024 × 1024 px; zoom: 0.75×; line averaging: 1; acquisition rate:400 Hz. We used ×63 oil immersion objective to capture images of dendritic spines. The secondary basal dendrites were selected by minimum of 50 μm and maximum of 110 μm from the soma. For each dendritic spine, z-stacks were captured with the following parameters: z-step: 0.15 μm; image size: 1024 × 1024 px; zoom: 3×; line averaging: 4; acquisition rate: 400 Hz. The pixel sizes for x-y are 0.06 μm.

We used the Reconstruct software[74] (v1.1.0.0) to analyze different types of spines by the following criteria[75]: mushroom-like spine (head diameter > 0.6 μm), thin spine (length > 1 μm), and stubby spine (ratio of spine length to the width of spine base < 1). Apical and basal dendrites are traced by Simple Neurite Tracer, a plugin available from image J (v1.53c). Sholl analysis characterized the number of intersections using 10 μm concentric shells. The investigators were blinded to the genotypes and treatment.

**Electrophysiology**. Rats were anesthetized by intraperitoneal injection of pentobarbital sodium (40 mg/kg) and perfused transcardially for 1 min with 4 °C modified artificial cerebrospinal fluid (aCSF) containing (in mM) 250 glycerol, 2 KCl, 10 MgSO$_4$, 0.2 CaCl$_2$, 1.3 NaH$_2$PO$_4$, 26 NaHCO$_3$, and 10 glucose, to protect CNS neurons and maintain functional connectivity of brain slices. Rats were then decapitated and brains were quickly removed and chilled in ice-cold ACSF for an additional 1 min. Transverse brain slices (350 μm in thickness) were prepared using a Vibroslice (VT 1000 S; Leica) in ice-cold ACSF. Slices were then incubated in regular ASCF containing (in mM): 126 NaCl, 3 KCl, 1.25 NaH$_2$PO$_4$, 1.0 MgSO$_4$, 2.0 CaCl$_2$, 26 NaHCO$_3$, and 10 glucose for 30 min at 34 °C for recovery, and then at room temperature (25 ± 1 °C) for an additional 2–8 h. All solutions were saturated with 95% O$_2$/5% CO$_2$ (vol/vol).

Whole-cell patch-clamp recordings from layer 5 pyramidal neurons in ACC were visualized with infrared optics using an upright microscope equipped with a ×40 water-immersion lens (BX51WI; Olympus) and infrared-sensitive CCD camera. All data were obtained with a HEKA EPC10 double patch-clamp amplifier. Data were low-pass filtered at 10 kHz and digitally sampled at 10 kHz with PatchMaster version 2 × 90.5. To record mEPSCs, we block GABAA receptor with 20 μM bicuculline methodiod (BMI) and action potentials with 1 μM TTX in ACSF, respectively. The pipettes (input resistance: 3–6 MΩ) solution contained (mM): 105 K-gluconate, 30 KCl, 10 HEPES, 10 phosphocreatine, 4 ATP-Mg, 0.3 GTP-Na, 0.3 EGTA, 5 QX-314 (pH 7.35, 285 mOsm). mIPSCs were recorded at holding potentials of +10 mV, 105 K-gluconate and 30 KCl was replace by 125 CsCH$_3$SO$_3$, 5 CsCl and 1 MgCl$_2$ in pipette solution. Twenty micromolar CNQX,

50 μM AP-5, and 1 μM TTX was added in the bath solution. The action potential was recorded by the current-clamp. The pipettes solution contained (mM): 135 K-gluconate, 7 KCl, 10 HEPES, 10 phosphocreatine, 4 ATP-Mg, 0.4 GTP-Na, 0.5 EGTA (pH 7.30, 285 mOsm). Neurons were held at −70 mV and were injected with different currents (duration, 500 ms; increments, 20 pA; from −200 to 400 pA; interval, 10 s). The input–output relationship was defined as the number of action potentials versus the amplitude of current injection. For recording spontaneous synaptic currents, pipettes were filled with cesium-based internal fluid (in mM: 125 CsCH$_3$SO$_3$, 5 CsCl, 1 MgCl$_2$ 10 HEPES, 4 Mg-ATP, 0.3 Tris-GTP, 10 Phosphocreatine, 5 QX-314). sEPSCs were recorded at a holding potential of −60 mV and sIPSCs were recorded at a holding potential of +10 mV. Spontaneous and miniature events were analyzed using Mini Analysis Program (Synaptosoft).

To record LTD from layer 5 pyramidal neurons in ACC, 4–5-week-old rats were anesthetized and decapitated. The brain was rapidly removed and placed in ice-cold (2–4 °C) oxygenated (95% O$_2$–5% CO$_2$) aCSF containing (in mM): 124 NaCl, 3 KCl, 26 NaHCO$_3$, 1.25 NaH$_2$PO$_4$, 1MgSO$_4$, 10 D-glucose, and 2 CaCl$_2$. The brain slices were cut at thickness of 400 μm using a vibratome before storing in room temperature aCSF for ≥1 h before use. A cesium-based solution (in mM: 135 CsCH3SO$_3$, 8 NaCl, 10 Hepes, 0.5 EGTA, 4 Mg-ATP, 0.3 Na-GTP, 10 phosphocreatine, 5 QX-314) was used to record evoked EPSC at room temperature. Cells in which series resistance exceeded 30 MΩ or changed >20% were discarded. A concentric bipolar stimulating electrode (CBAPB50; FHC) was placed on the layer 2/3, and synaptic responses were evoked using 0.1 ms constant-current pulses. Picrotoxin (50 μM) and NBQX (5 μM)/AP-5(50 μM) were bath applied and cells were held at −40 mV/−70 mV to isolate EPSC$_{NMDA}$/ EPSC$_{AMPA}$. Activity-dependent LTD was induced by delivery of TFS (300 pulses at 5 Hz) at baseline stimulus intensity. The magnitude of LTD was calculated at the amplitudes of NMDA currents 50–55 min after TFS.

**Stereotaxic injection of AAV into ACC**. The pAAV2/9-EF1α-loxp-stop-loxp-tdTomato-WPRE-poly A was generated by Obio Technology (Shanghai) Corp., Ltd. The titer of this AAV is 10$^{13}$/μl and we injected 0.5 μl AAV into each side of ACC. Five-week-old male Drd2-Cre rats were anesthetized by intraperitoneal injection of pentobarbital sodium (40 mg/kg) and head-fixed in a stereotaxic device (RWD life science). The AAV was delivered through stereotaxic injector (Stoelting) with a speed of 0.1 μl / min. The injection coordinates are as follows: anteroposterior (AP) +1.55 mm, dorsoventral (DV) −2.25 mm, mediolateral (ML) ± 0.55 mm relative to Bregma.

The pAAV2/9-EF1α-loxp-EGFP-Drd2 or control shRNA-loxp was generated by Taitool Bioscience (Shanghai) Corp., Ltd. We used an shRNA expressing system called PRIMER (potent RNA interference using microRNA expressing vectors)[76], which allows for the multicistronic cotranscription of a reporter gene, thereby

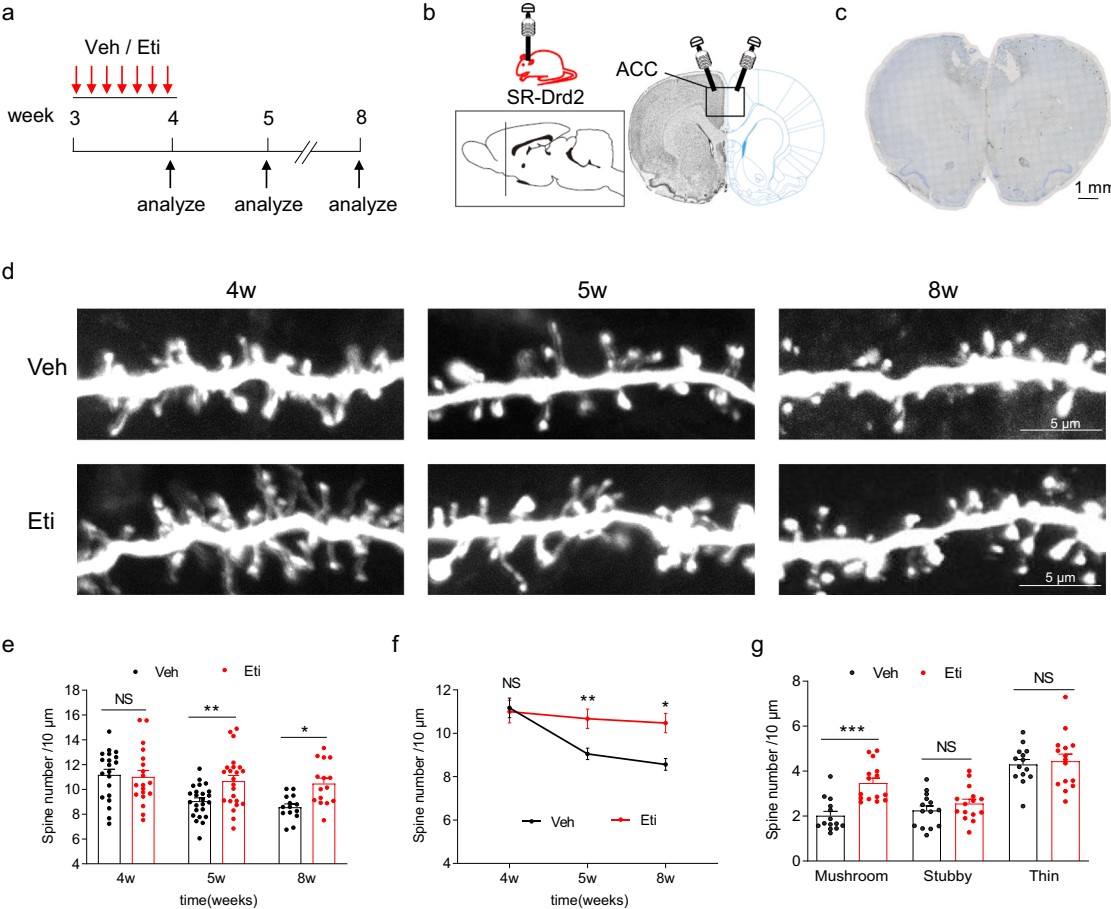

**Fig. 7 Impaired synaptic pruning by DRD2 inhibition during adolescence. a** Experimental design. SR-Drd2 rats received daily injection of Eti (1 µg in 0.5 µl per side) or Veh (0.5 µl saline per side) into layer 5 of ACC between 3- and 4-week-old age, and the spine densities of secondary basal dendrites in Drd2-positive neurons were analyzed at different timepoints. **b** Diagram showing the injection of Veh or Eti into layer 5 of ACC in SR-Drd2 rat. The dashed line of the sagittal section diagram indicates the position of the coronal section. The rectangle indicates the brain regions of ACC. **c** Nissl staining to determine the injection sites. Scale bar, 1 mm. **d** Representative dendritic spines of Drd2-positive neurons at different aged SR-Drd2 rats, as in panel **a**. Scale bar, 5 µm. **e** Impaired synaptic pruning in Drd2-postive neurons when inhibition of Drd2 between 3- and 4-week-old age. The spine densities of Drd2-positive neurons from different aged SR-Drd2 rats treated with Veh or Eti were quantified. NS not significant, $P = 0.9879$, two-way ANOVA followed by Sidak's multiple comparisons test, $n = 20$ dendrites from 3 control rats, $n = 19$ dendrites from 3 Eti-treated rats. Data are presented as mean values ± SEM. **$P = 0.0091$, two-way ANOVA followed by Sidak's multiple comparisons test, $n = 24$ dendrites from 4 control rats, $n = 23$ dendrites from 4 Eti-treated rats. Data are presented as mean values ± SEM. *$P = 0.0181$, two-way ANOVA followed by Sidak's multiple comparisons test, $n = 14$ dendrites from 3 control rats, $n = 15$ dendrites from 3 Eti-treated rats. Data are presented as mean values ± SEM. **f** Line chart of the data in panel **e**. The statistical results are same as panel **e**. **g** Increased densities of mushroom-like spines in Drd2-positive neurons after inhibition of DRD2 between 3- and 4-week-old age. The densities of different types of dendritic spines in Drd2-positive neurons from 8-week-old rats were quantified. ***$P < 0.0001$, NS not significant, $P$ (stubby) $= 0.7185$, $P$ (thin) $= 0.9601$, two-way-ANOVA followed by Sidak's multiple comparisons test, $n = 14$ dendrites from 3 control rats, $n = 15$ dendrites from 3 Eti-treated SR-Drd2 rats. Data are presented as mean values ± SEM.

facilitating the tracking of shRNA production in individual cells. The target sequence of Drd2 shRNA against rat Drd2 is as follows ACTCAGATGCTTGCCA TTGTT. The control shRNA contains a nontargeting sequence as follows AAATGTACTGCGCGTGGAGAC. The titer of this AAV is $10^{13}$/µl and we injected 0.5 µl AAV into each side of ACC. Three-week-old male Drd2-Cre rats were anesthetized by intraperitoneal injection of pentobarbital sodium (40 mg/kg) and head-fixed in a stereotaxic device (RWD life science). The AAV was delivered through stereotaxic injector (Stoelting) with a speed of 0.1 µl / min. The injection coordinates are as follows: anteroposterior (AP) + 1.40 mm, dorsoventral (DV) −1.95 mm, mediolateral (ML) ± 0.47 mm relative to Bregma.

**Microinjection of Eti and Rap or Veh into layer 5 of ACC.** Rats were anesthetized by intraperitoneal injection of pentobarbital sodium (40 mg/kg) and head-fixed in a stereotaxic device (RWD life science). In ACC infusion experiments, we tilt 20° in canula surgery to avoid collision. Rats were implanted bilaterally with stainless steel guide cannula (RWD life science). Adolescent rat coordinates were as follows: anteroposterior (AP)+ 1.40 mm, dorsoventral (DV)− 1.10 mm, mediolateral (ML) ± 1.20 mm relative to Bregma. Adult rat coordinates were as follows: ante- roposterior (AP)+ 1.70 mm, dorsoventral (DV)− 1.70 mm, mediolateral (ML) ±

1.50 mm relative to Bregma. Internal cannula extending 1 mm beyond the guides were used for drug infusions. We used infusion pump to perform daily micro- injection of Eti (Sigma, E101, 1 µg in 0.5 µl per side), Rap (Selleck Chemicals, S1039, 1 µM in 0.5 µl per side), or Veh (0.5 µl saline per side for Eti, 0.5 µl DMSO per side for Rap,) into deep layers of ACC. The dosage of Eti and Rap we used has been shown to be able to selectively inhibit DRD2 and mTOR in rat ACC, respectively[43,50].

**Behavior analysis**
*Open field.* Adult male rats were placed in a chamber (42 × 42 × 37 cm) and monitored for movement for 30 min. The total distance of open field was measured by the True-Scan System (Coulbourn Instruments). The chamber was cleaned with 75% ethanol and dried thoroughly after each test session.

*Light-dark box.* Adult male rats were placed in a box (42 × 42 × 37 cm), the half of left box was light, and the other part was dark. Rats were monitored for 10 min. The total time staying in light and dark box was measured by the True-Scan System (Coulbourn Instruments). The box was cleaned with 75% ethanol and dried thoroughly after each test session.

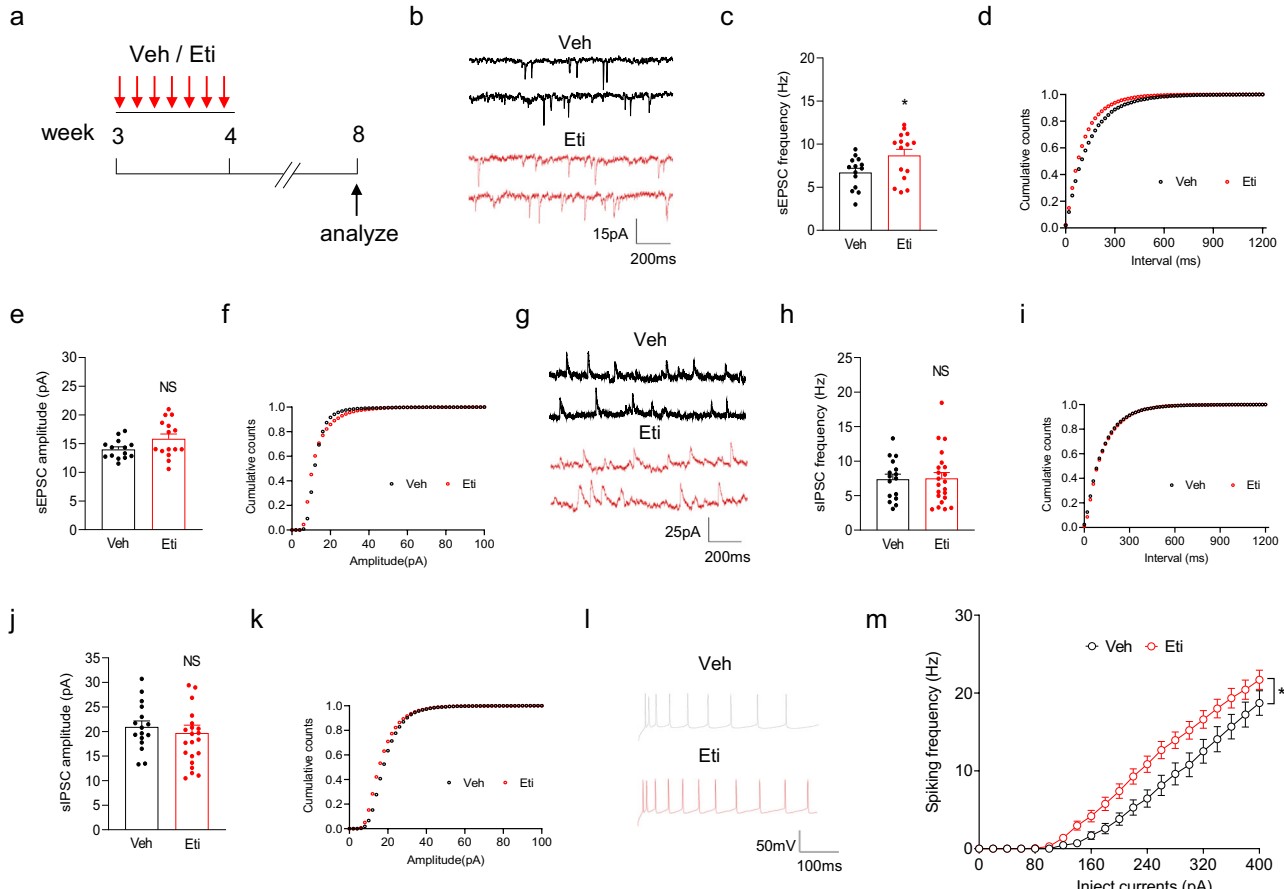

**Fig. 8 Hyperglutamatergic function of Drd2-positive neurons after inhibition of DRD2 during adolescence. a** Experimental design. SR-Drd2 rats received daily injection of Eti (1 μg in 0.5 μl per side) or Veh (0.5 μl saline per side) into layer 5 of ACC between 3- and 4-week-old age, the sEPSC, sIPSC, and cell excitability of Drd2-positive neurons were then analyzed at 8-week-old age. **b** Representative sEPSC traces of Drd2-positive neurons from 8-week-old Eti and Veh-treated rats. **c** Increased sEPSC frequency of Drd2-positive neurons in Eti-treated SR-Drd2 rats, compared with controls. *P = 0.0324, two-sided t-test, n = 14 neurons from 3 control rats, n = 15 neurons from 3 Eti-treated rats. Data are presented as mean values ± SEM. **d** Cumulative plots of sEPSC frequency. **e** Similar sEPSC amplitude of Drd2-positive neurons between Veh and Eti- treated rats. NS not significant, P = 0.0672, two-sided t-test, n = 14 neurons from 3 control rats, n = 15 neurons from 3 Eti-treated rats. Data are presented as mean values ± SEM. **f** Cumulative plots of sEPSC amplitude. **g** Representative sIPSC traces of Drd2-positive neurons from 8-week-old Veh and Eti-treated rats. **h** Similar sIPSC frequency of Drd2-positive neurons between Veh and Eti-treated rats. NS not significant, P = 0.9176, two-sided t-test, n = 16 neurons from 3 control rats, n = 22 neurons from 3 Eti-treated rats. Data are presented as mean values ± SEM. **i** Cumulative plots of sIPSC frequency. **j** Similar sIPSC amplitude of Drd2-positive neurons between Veh and Eti-treated rats. NS not significant, P = 0.5628, two-sided t-test, n = 16 neurons from 3 control rats, n = 22 neurons from 3 Eti-treated rats. Data are presented as mean values ± SEM. **k** Cumulative plots of sIPSC amplitude. **l** Representative action potentials of Drd2-positive neurons from 8-week-old Veh and Eti-treated rats. **m** Increased excitability of Drd2-positive neurons in Eti-treated rats, compared with controls. Shown are the I/O curves of action potentials from Drd2-positive neurons. * Genotype $F_{(1, 62)}$ = 5.933, P = 0.0177, two-way ANOVA, n = 31 neurons from 4 control rats, n = 33 neurons from 4 Eti-treated rats. Data are presented as mean values ± SEM.

*Elevated O-maze.* The elevated O-maze is an annular platform with a width of 5.5 cm and an outer diameter of 92 cm. The elevated maze has two open parts and two opposite closed parts. While the closed parts were enclosed by side walls of 20 cm height, there were no walls for the open parts. The apparatus was elevated 50 cm above the floor. During the test, the rats were placed into one of the open parts facing the closed part, and were allowed to freely explore the maze for 10 min. The behavior of each rat was recorded by a digital camera and was analyzed using the Any-maze system (Stoelting). The maze was wiped clean with 75% ethanol after each session.

*Social interaction and novelty.* Adult male rats were tested for social behavior in a three-chamber box (70 × 30 × 35 cm). Each of the end chambers contains a clear Plexiglas cylinder. One cylinder is the "social" cylinder, which contains a stimulus rat (adult wild-type male rat who never met test rat). The other cylinder is "non-social" cylinder, which is empty. The test rats were first placed in the center chamber and allowed to freely explore the chambers with two empty cylinders for 10 min. For the social interaction test, rats were given an additional 10 min to explore the chambers with a "social" cylinder (S1) and a "non-social" cylinder (O). Thirty minutes after the social interaction test, the rats were subjected to a social

novelty test. The test rats were given an additional 10 min to explore the chambers with cylinders containing a familiar rat (S1) and a novel rat (S2). Six-week-old male SD rats were used as social opponents. Sessions were video-recorded, and time spent around the "social" cylinder and "non-social" cylinder, or the familiar cylinder and novel cylinder were analyzed by the ANY-maze video tracking system (Stoelting). The box and cylinder were cleaned with 75% ethanol and dried thoroughly after each test session.

**Statistical analysis**. Two-way ANOVA was used in Sholl analysis, spine analysis at different timepoints, behavioral analysis including open field and social interaction, electrophysiological studies including I/O curve of AP. One-way ANOVA was used for the analysis of the data from three or more groups. Student's t-test was used to compare data from two groups. Data were expressed as mean ± SEM unless otherwise indicated. The sample size justification was based on the previous studies[77]. The outliers were calculated with the Grubbs's test. The example data shown were close to the overall mean. Statistically significant difference was indicated as follows: ***P < 0.001, **P < 0.01, and *P < 0.05. The statistical analysis was performed with the software of GraphPad Prism 8.

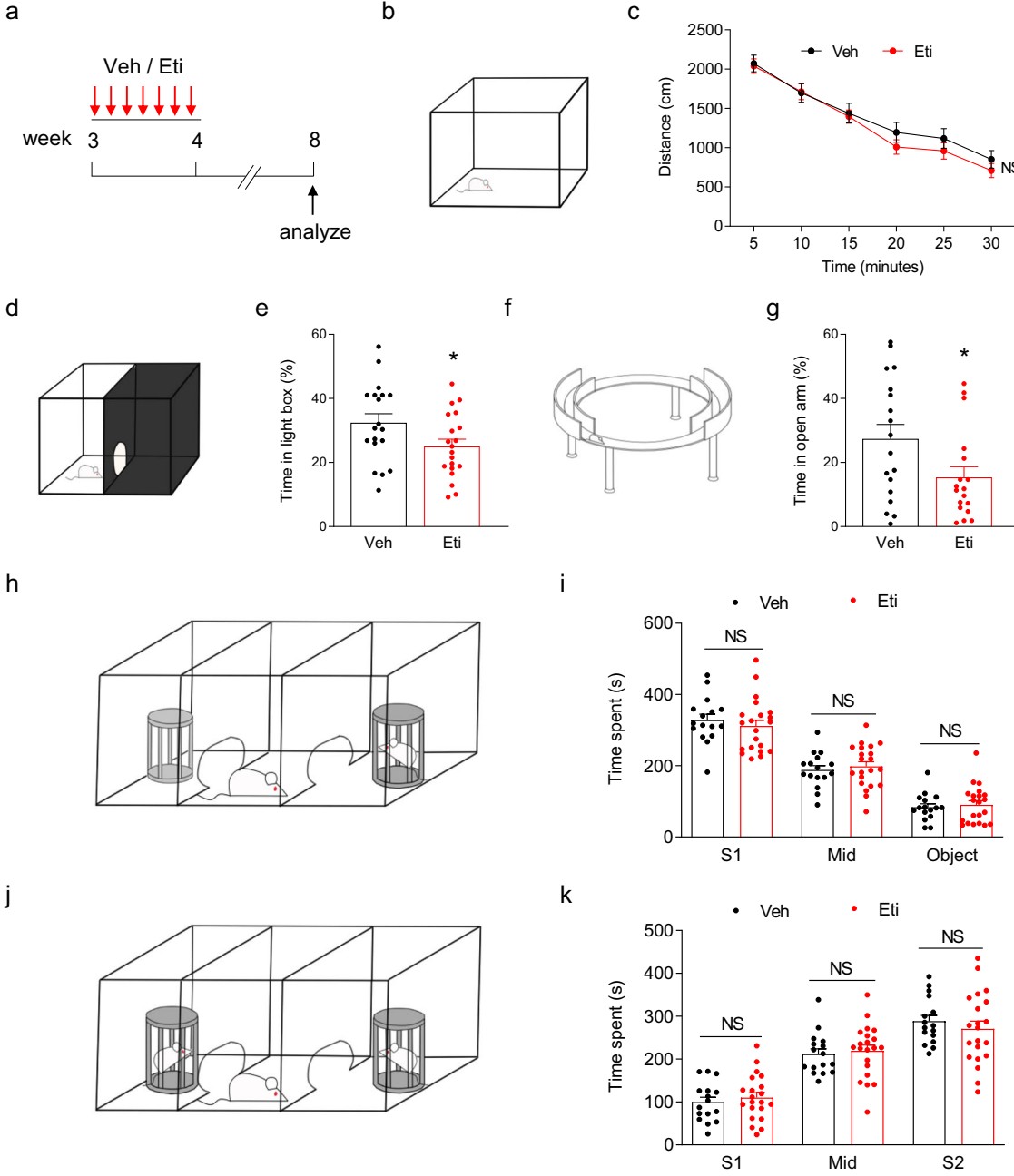

**Fig. 9 Elevated anxiety-like behavior after inhibition of DRD2 during adolescence. a** Experimental design. WT rats received daily injection of Eti (1 μg in 0.5 μl per side) or Veh (0.5 μl saline per side) into layer 5 of ACC between 3- and 4-week-old age, and behaviors were analyzed at 8-week-old age. **b** Schematic diagram of open field. **c** Travel distance in open field was no difference between Eti and Veh-treated rats. NS not significant, Genotype F (1, 246) = 2.133, P = 0.1454, two-way ANOVA, n = 22 for controls, n = 21 for Eti-treated rats. Data are presented as mean values ± SEM. **d** Schematic diagram of light-dark box. **e** Time spent in light box was reduced in Eti-treated rats, compared with controls. *P = 0.046, two-sided t-test, n = 19 for controls, n = 20 for Eti-treated rats. Data are presented as mean values ± SEM. **f** Schematic diagram of elevated maze. **g** Time spent in open arm was decreased in Eti-treated rats, compared with controls. *P = 0.0376, two-sided t-test, n = 18 for controls, n = 18 for Eti-treated rats. Data are presented as mean values ± SEM. **h** Schematic diagram of three-chamber test to study social interaction. **i** Social interaction was normal in Eti-treated rats. Time spent in each chamber was quantified. NS not significant, P (S1) = 0.7722, P (Mid) = 0.9379, P (Object) = 0.9806, two-way ANOVA followed by Sidak's multiple comparisons test, n = 16 for controls, n = 21 for Eti-treated rats. Data are presented as mean values ± SEM. **j** Schematic diagram of three-chamber test to study social novelty. **k** Social novelty was intact in Eti-treated rats. Time spent in each chamber was quantified. NS not significant, P (S1) = 0.9369, P (Mid) = 0.9765, P (S2) = 0.7556, two-way ANOVA followed by Sidak's multiple comparisons test, n = 16 for controls, n = 21 for Eti-treated rats. Data are presented as mean values ± SEM.

**Reporting summary**. Further information on research design is available in the Nature Research Reporting Summary linked to this article.

## Data availability

All data supporting the results presented herein are available from the article paper, supplementary information, and source data. The full-length images for all the gels or blots are provided in Supplementary Fig. 9. Source data are provided with this paper.

## Material availability

All unique materials used are readily available from the corresponding author upon request.

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

## Acknowledgements

This work was supported by the grant from National Natural Science Foundation of China (31861143033); the Fundamental Research Funds for the Central Universities, the grants from State Key Laboratory of Neuroscience and Shanghai Key Laboratory of Psychotic Disorders (No. 13dz2260500). D.-M.Y. is a NARSAD Young Investigator. We thank Dr. Dali Li in ECNU for helping to generate Drd2$^{+/-}$ rats.

## Author contributions

Y.-Q.Z. and L.-P.H. performed biochemical and behavioral experiments. W.-P.L. and B.Z. performed the electrophysiology recordings. C.-C.Z. performed the morphological analysis. D.-M.Y. designed the experiments, supervised the work, and wrote the paper.

## Competing interests

The authors declare no competing interests.

## Additional information

**Peer review information** *Nature Communications* thanks David Sulzer and the other anonymous reviewer(s) for their contribution to the peer review this work. Peer reviewer reports are available.

