## [Peer Review File · Nature Communications]

Reviewers' Comments:

Reviewer #1:

Remarks to the Author:

In this manuscript, Zhang et al. present compelling new data pointing towards a role for dopamine D2 receptor (D2R) signaling in cortical synaptic pruning during the late adolescent period. Either developmental genetic reduction in D2R expression levels or pharmacological D2R blockade during an apparent sensitive period in adolescence resulted in persistence of dendritic spines and reduction in NMDAR LTD compared to controls. The role of D2R in the cortex, especially as related to postnatal development and disease etiology, has been understudied. However, the mechanistic insight afforded by this study is quite overstated, and the hypotheses are somewhat underdeveloped.

Major:

1.) The central hesitation with this manuscript is the overstatement of any mechanistic understanding of D2R modulation of synaptic pruning illuminated by this study. The authors convincingly show that disruption of D2R signaling results in absence of developmental reduction in spine density and relative dysfunction in NMDAR LTD. However, there is no further exploration of how signaling through D2R effects these developmental steps.

a. It has been proposed that LTD onto some pyramidal populations is mediated by presynaptic D2Rs (Rocchetti et al. 2015) (<https://doi.org/10.1016/j.biopsych.2014.03.013>) What are the relative roles of acute D2R activation versus chronic stimulation in D2-mediation of LTD phenotype? (i.e., is NMDA LTD in ACC acutely D2-mediated?)

b. What are potential mechanisms by which signal transduction through D2R may mediate pruning? Is there evidence pointing towards a potential mechanism?

2.) Throughout the paper, the authors make claims to a cell autonomous mechanism, but this is overstated, and including the experiments on miniature potentials, they do not have clear evidence that, for example, the changes are not driven presynaptically. D2R expression of dopamine autoreceptors may have major effects on dopamine release. It is possible that the change in DA release could mediate the behavioral findings or even the developmental trajectory in spine density.

Presumably the model rat has reduction in D2R expression globally, including on presynaptic DA terminals. As the authors are introducing a new genetic model, although the study is on cortical synapses, they should acknowledge this and show the changes in D2R that could impinge on the circuit, including if D2R is markedly changed or not on midbrain DA neurons and striatal neurons.

Minor:

1.) As mentioned, a formulation of the phrase "...synaptic pruning through cell-autonomous mechanisms sharing with long-term depotentiation..." is repeated many times (lines 55, 111, 126, 232, 254, 317, and 344). The point is taken that there is likely commonality between the cellular/biochemical processes underlying these phenomena, but they are not identical and there are critical differences in time scale, mechanism, etc. We are confused by the focus on this point, and question whether it adds to the message of the manuscript. In addition, the syntax of several of these statements (e.g. "...mechanisms sharing with LTD..." ; "...share the mechanisms with synaptic pruning") are confusing/ungrammatical.

2.) That antipsychotics target the D2R is mentioned as relevant to the impact of these findings a few times (L.50, 366). However, since these drugs are D2R antagonists, shouldn't they be hypothesized to exacerbate symptomatology based on these findings?

3.) "Sensitive period" may be more precise/appropriate for the findings rather than "critical time window."

4.) "Hyperglutamatergic" should be a single word or hyphenated rather than two words (lines 57, 205, 229, 319...)

5.) L.302 please identify within the text the type of elevated maze used for anxiety behavior.

6.) L.315-316 please explain what is meant by "...the control and SR-Drd2+/- rats" (too abbreviated/colloquial). I would remove the term "self-reporting" as it is a typical reporter of transfection.

7.) Please include detail on immunofluorescence quantification methodology.

8.) Fig. 1C-D: In the chosen image, it looks as if there are many more layer 6 Drd2+ cells than layer 2/3. Is this image representative of the summary data? How were bounds of ACC defined for cell counts?

9.) Fig. 3Q-R; L.221-227: It is suggested here that the excitability implied by greater spine density should be directly related to excitability given direct depolarizing current injection. This is imprecise, and the findings here actually support greater cell-intrinsic excitability above and beyond that driven by enhanced synaptic input.

Reviewer #2:

Remarks to the Author:

In this original article submitted to Nature Communications, Zhang and colleagues show the role of Dopamine D2 receptors in synaptic pruning in the frontal cortex in rats. Using complementary technical approaches, the authors successfully identified D2 positive neurons and manipulated D2R expression in the ACC to demonstrate a critical role of the D2R in excitatory synapses elimination occurring during late adolescence in rats. The structural plasticity impact is associated with electrophysiological differences such as an increased excitability of the D2R +/- neurons as well as an increase in the miniature EPSC frequency. The authors then hypothesized that the effect may be linked to changes in LTD mechanisms and using TFS in the layer 2/3 and measure of the Peak amplitude in patched layer5 pyramidal neurons. Remarkably, the spine density and electrophysiological effects induced by D2R downregulation are mimicked using Eticlopride, a D2R antagonist. Finally, injection of this drug in the ACC induces anxiety-like behaviors.

Strengths:

I deeply appreciate the quality of the experiments and the large panel of methods used to support the main messages of the paper. The specific time window of the effect of the d2r downregulation is really interesting at the fundamental level but also for a better understanding of psychiatric diseases.

The results are well illustrated and explained and the stats are adequate with a good sample size for neurons and animals.

However, a few points need to be corrected for this article to be published on Nature communications.

Weaknesses and points to correct :

In Methods:

1)The pixel sizes in microns for X-Y are missing.

2) The authors need to better explain their methods of classification for the mushroom, thin and stubby spines and give in supplementary figures different examples of each category.

3) In the case of subjective classification, it is required to show as well objective spine measurements: length and spine head size, to be transparent and for other labs to reproduce it.

In Results:

4) The idea that interneurons express less D2R than pyramidal cells seems a bit over-interpreted. Interneurons represent on average around 20% of the cortical neurons, it makes sense to me that D2+ interneurons represent only 9/10% of the total, which is still almost half of the interneurons. The authors chose to study pyramidal cells and it is fine, but telling that D2R is mainly expressed in pyramidal cells is misleading considering the importance of interneurons for psychiatric disorders and especially for schizophrenia. Also, interneurons are major regulators of neuronal

network activity, so the effects on behaviors might still be dependent on them. I propose that the authors adapt their justification of studying pyramidal neurons to be more objective.

5) The authors have a really good genetic model of D2R downregulation, so, using Eticlopride for behavioral studies, seems a bit surprising. It would be interesting to know if the behavioral effects of genetic D2R downregulation are the same as the effects of the drug.

6) In figure 2.d the GAPDH is overexposed, an exposure similar to 2.c would allow a proper quantification.

7) In figure 2,3,4 some of the dendrites showed in examples are blurry, probably due to a movement of the samples during the z-stack. I would like the authors to find better illustrative neurons

Discussion:

8) In the 2nd paragraph, "Drd2 increased dendritic spines in hippocampal pyramidal", a term is missing: "density"

9) There is still a chance that behavioral effects are due to changes in presynaptic dopamine release through presynaptic D2R projecting into the ACC, independently of the synaptic pruning, this aspect can be discussed as well.

10) D2+/- neurons are hyperexcitable and the authors showed it nicely, however, this effect might also be dependent on intrinsic excitability due to changes in ion channel properties as well, for example, due to cAMP and HCN1 or even due to effects on Voltage-gated channels. The authors could discuss those aspects as well.

Reviewer #3:

Remarks to the Author:

In their paper entitled "Dopamine D2 receptor regulates cortical synaptic pruning" Ya-Qiang Zhang et al., investigated the role of the dopamine receptor 2 (Drd2) in synaptic pruning during adolescence in a self-reporting Drd2 heterozygous rat. They found that Drd2 may play important roles in synaptic pruning during the transition from adolescence to adulthood without effects on spine formation. They demonstrated that the role of Drd2 in NMDAR-mediated LTD is cell-autonomous and that inhibition of Drd2 during adolescence may increase glutamatergic transmission in adults. This has important implications in synaptic plasticity at crucial developmental stages, and thus they highlight the potential role of Drd2 in development of anxiety.

The flow of the paper was good, and I want to compliment the diagrams in the figures that offer a visual explanation of the experiments performed.

However, there are some comments that I would like the authors to address before recommending the paper for publication.

Minor comments:

- I suggest that authors adding the age of the rats when referred to periods like "adolescence" and "adulthood" to remind the reader the time points used for each developmental stage (for example, in line 160, line 304, etc...)

- In line 201 stubby and mushroom-shaped spines are grouped as mature spines. Other researchers have considered stubby morphologies as immature or transitory, rather than mature (see references below*). Why do authors consider stubby morphologies mature in this work? Can the authors provide other references that indicate this classification as accepted?

*See:

Harris KM, Jensen FE, Tsao B. Three-dimensional structure of dendritic spines and synapses in rat hippocampus (CA1) at postnatal day 15 and adult ages: implications for the maturation of synaptic

physiology and long-term potentiation. *J Neurosci.* 1992 Jul;12(7):2685-705. doi: 10.1523/JNEUROSCI.12-07-02685.1992. PMID: 1613552; PMCID: PMC6575840.
Berry KP, Nedivi E. Spine Dynamics: Are They All the Same? *Neuron.* 2017 Sep 27;96(1):43-55. doi: 10.1016/j.neuron.2017.08.008. PMID: 28957675; PMCID: PMC5661952.
Pchitskaya E, Bezprozvanny I. Dendritic Spines Shape Analysis-Classification or Clusterization? Perspective. *Front Synaptic Neurosci.* 2020 Sep 30;12:31. doi: 10.3389/fnsyn.2020.00031. PMID: 33117142; PMCID: PMC7561369

- In line 243 authors mention that *Drd2* is important in NMDAR-mediated LTD. Other groups have looked into this and should be cited. (e.g. Sheynikhovich et al., *J Neuroscience* 2013).

- I suggest to include the age of the rats used for each experiment in the methods section to have all the information handy for the reader.

Major comments:

- Authors should explain the criteria used to classify spines in the different categories (mushroom, thin and stubby). If this classification was subjective to the investigator analyzing the images, I strongly suggest to repeat the classification using an unbiased method. There is available (cost-free) software such as NeuronStudio software (Computational Neurobiology and Imaging Center CNIC, Mount Sinai School of Medicine, NY RRID:SCR_005793) or Reconstruct (Fiala, J.C. (2005), Reconstruct: a free editor for serial section microscopy. *Journal of Microscopy*, 218: 52-61. <https://doi.org/10.1111/j.1365-2818.2005.01466.x>).

- In the Statistical Analysis of the Methods section the authors quote the Wikipedia as the source to decide on their approach to eliminate outliers from their datasets. The authors don't explain properly why they decided that eliminating outliers was necessary. Outliers should be calculated with a proper statistical test such as Grubbs's test. If a big section of the datasets were eliminated through their formula, the reality may be that there are no significant effects. If the biological variability was too wide, normalizing data would be more appropriate than eliminating data points. Authors should provide complete datasets including the outliers that have been removed, and explain why they deemed necessary to calculate and eliminate outliers.

Reviewer #1 (Remarks to the Author):

In this manuscript, Zhang et al. present compelling new data pointing towards a role for dopamine D2 receptor (D2R) signaling in cortical synaptic pruning during the late adolescent period. Either developmental genetic reduction in D2R expression levels or pharmacological D2R blockade during an apparent sensitive period in adolescence resulted in persistence of dendritic spines and reduction in NMDAR LTD compared to controls. The role of D2R in the cortex, especially as related to postnatal development and disease etiology, has been understudied. However, the mechanistic insight afforded by this study is quite overstated, and the hypotheses are somewhat underdeveloped.

Response – We thank this reviewer for his/her comments that “Zhang et al. present compelling new data pointing towards a role of D2R signaling in cortical synaptic pruning...”. We are also grateful for the constructive critiques which will significantly improve the manuscript.

Major:

1.) The central hesitation with this manuscript is the overstatement of any mechanistic understanding of D2R modulation of synaptic pruning illuminated by this study. The authors convincingly show that disruption of D2R signaling results in absence of developmental reduction in spine density and relative dysfunction in NMDAR LTD. However, there is no further exploration of how signaling through D2R effects these developmental steps.

a. It has been proposed that LTD onto some pyramidal populations is mediated by presynaptic D2Rs (Rocchetti et al. 2015) (<https://doi.org/10.1016/j.biopsych.2014.03.013>) What are the relative roles of acute D2R activation versus chronic stimulation in D2-mediation of LTD phenotype? (i.e., is NMDA LTD in ACC acutely D2-mediated?)

Response – We performed additional experiments to address whether NMDA LTD in ACC is acutely regulated by Drd2. We treated the ACC slices from 4-week-old Drd2 reporter rats with vehicle or Eti, a Drd2 antagonist for 20 min and then recorded NMDA LTD on pyramidal neurons in layer 5 of ACC. The newly obtained results indicated that acute inhibition of Drd2 impaired NMDA LTD in Drd2-positive but not Drd2-negative neurons (Fig. 6d and e).

Unlike the brain slices where mesohippocampal dopamine pathway was left intact (Rocchetti et al., 2015), the brain slices used in our study did not contain the dopaminergic neurons which were disconnected with ACC during the preparation of brain slices. On the other hand, the cellular expression pattern of Drd2 is different between hippocampus and ACC. Drd2 is not expressed postsynaptically in CA1 pyramidal neurons in the hippocampus (Puighermanal et al., 2015; Rocchetti et al., 2015; Yu et al., 2019), while Drd2 is highly expressed in layer 5 pyramidal neurons in the ACC. Taken together, these results suggest that D2 regulates LTD through

different mechanisms in the hippocampus and ACC. We discussed these points and cited the references in the revised manuscript (line 427-436).

b. What are potential mechanisms by which signal transduction through D2R may mediate pruning? Is there evidence pointing towards a potential mechanism?

Response – The downstream of DRD2, a G_{i/o}-coupled receptor, includes cAMP-PKA and β-arrestin-PP2A-AKT signaling (Beaulieu et al., 2005). Downregulation of DRD2 has been shown to activate AKT-mTOR (mammalian target of rapamycin) signaling (Beaulieu et al., 2007; Bonito-Oliva et al., 2013), and activation of mTOR signaling has been implicated in the deficient synaptic pruning in the autism-spectrum diseases (Tang et al., 2014). Due to these facts, we hypothesized activation of mTOR signaling might be also involved in the impaired synaptic pruning in *Drd2*^{+/-} rats.

We performed additional experiments to address whether activation of mTOR signaling during adolescence is important for the deficient synaptic pruning in the *Drd2*^{+/-} rats. The newly obtained results indicated that AKT-mTOR signaling was increased in the ACC from 3-4-week-old *Drd2*^{+/-} rats, compared with controls (Fig. 4b-h). Moreover, administration of rapamycin, the mTOR inhibitor into the ACC during 3-4-week-old age rescued the abnormal spine density and glutamatergic transmission in adult *Drd2*^{+/-} rats (Fig. 4i-s). These results provide evidence that elevated mTOR signaling might represent the potential mechanism underlying the impaired synaptic pruning in the *Drd2*^{+/-} rats.

2.) Throughout the paper, the authors make claims to a cell autonomous mechanism, but this is overstated, and including the experiments on miniature potentials, they do not have clear evidence that, for example, the changes are not driven presynaptically. D2R expression of dopamine autoreceptors may have major effects on dopamine release. It is possible that the change in DA release could mediate the behavioral findings or even the developmental trajectory in spine density.

Presumably the model rat has reduction in D2R expression globally, including on presynaptic DA terminals. As the authors are introducing a new genetic model, although the study is on cortical synapses, they should acknowledge this and show the changes in D2R that could impinge on the circuit, including if D2R is markedly changed or not on midbrain DA neurons and striatal neurons.

Response – We agree with this reviewer that the *Drd2*^{+/-} rats have reduction in *Drd2* expression globally because they were heterozygous mutation. To demonstrate whether *Drd2* regulates synaptic pruning in a cell-autonomous way, we injected adeno-associated virus (AAV) expressing Cre-dependent control or *Drd2* miRNA and EGFP into the ACC of 3-week-old *Drd2*-Cre rats (Fig. 5a-d). The newly-obtained results indicated that downregulation of *Drd2* expression in the ACC during adolescence led to increased spine density, elevated glutamatergic transmission and heightened neuronal excitability of *Drd2*-positive neurons in adulthood (Fig. 5e-n), as

what we found in SR-Drd2^{+/-} rats. Since expression of Drd2 miRNA only occurred in the Drd2-positive neurons (i.e., Cre-dependent) in the ACC of Drd2-Cre rats, these newly-obtained results provide evidence that Drd2 regulates synaptic pruning through a cell-autonomous manner. Nonetheless, we agree with this reviewer that the behavioral deficits could be due to changes in presynaptic dopamine release through presynaptic D2R projecting into the ACC. We discussed this point in the revised manuscript (line 453-455).

Minor:

1.) *As mentioned, a formulation of the phrase "...synaptic pruning through cell-autonomous mechanisms sharing with long-term depotentiation..." is repeated many times (lines 55, 111, 126, 232, 254, 317, and 344). The point is taken that there is likely commonality between the cellular/biochemical processes underlying these phenomena, but they are not identical and there are critical differences in time scale, mechanism, etc. We are confused by the focus on this point, and question whether it adds to the message of the manuscript. In addition, the syntax of several of these statements (e.g. "...mechanisms sharing with LTD..."; "...share the mechanisms with synaptic pruning") are confusing/ungrammatical.*

Response – Sorry for being inaccurate. We agree with this reviewer that some similar cellular/biochemical processes might underly synaptic pruning and LTD, but they are not identical and have critical difference. We deleted the statement of "...synaptic pruning through cell-autonomous mechanisms sharing with long-term depotentiation...". Instead, we stated that "LTD included some similar cellular/biochemical processes with synaptic pruning" in the revised manuscript.

2.) *That antipsychotics target the D2R is mentioned as relevant to the impact of these findings a few times (L.50, 366). However, since these drugs are D2R antagonists, shouldn't they be hypothesized to exacerbate symptomatology based on these findings?*

Response – Sorry for being unclear. Schizophrenia is thought to be due to the excessive synaptic pruning in the cerebral cortex (Keshavan et al., 1994; Konopaske et al., 2014; Penzes et al., 2011). Here we show that Drd2 antagonist attenuated cortical synaptic pruning, which is hypothesized to be beneficial for schizophrenia patients. We discussed these points in the revised manuscript (line 464-467).

3.) *"Sensitive period" may be more precise/appropriate for the findings rather than "critical time window."*

Response – Agree. We replaced "critical time window" with "sensitive period" in the revised manuscript.

4.) *"Hyperglutamatergic" should be a single word or hyphenated rather than two*

words (lines 57, 205, 229, 319...)

Response – We used a single word of “hyper-glutamatergic” in the revised manuscript.

5.) L.302 please identify within the text the type of elevated maze used for anxiety behavior.

Response – We used elevated O maze for anxiety behavior (line 381).

6.) L.315-316 please explain what is meant by “...the control and SR-Drd2+/- rats” (too abbreviated/colloquial). I would remove the term “self-reporting” as it is a typical reporter of transfection.

Response – Agree. We removed the term “self-reporting” (line 394).

7.) Please include detail on immunofluorescence quantification methodology.

Response – We provided the website link to the detailed protocol of unbiased stereology to quantify the immunofluorescence in the revised manuscript (line 512-513).

8.) Fig. 1C-D: In the chosen image, it looks as if there are many more layer 6 Drd2+ cells than layer 2/3. Is this image representative of the summary data? How were bounds of ACC defined for cell counts?

Response – We reanalyzed the cell number in different layers of ACC (Fig. 1d). The bounds of rat ACC were defined following the previous publication (Lee et al., 2007) (line 513-514).

9.) Fig. 3Q-R; L.221-227: It is suggested here that the excitability implied by greater spine density should be directly related to excitability given direct depolarizing current injection. This is imprecise, and the findings here actually support greater cell-intrinsic excitability above and beyond that driven by enhanced synaptic input.

Response – We agree with this reviewer that data in Fig. 3q-r support the greater cell-intrinsic excitability. We stated as such in the revised manuscript.

Reviewer #2 (Remarks to the Author):

In this original article submitted to Nature Communications, Zhang and colleagues show the role of Dopamine D2 receptors in synaptic pruning in the frontal cortex in rats. Using complementary technical approaches, the authors successfully identified D2 positive neurons and manipulated D2R expression in the ACC to demonstrate a

critical role of the D2R in excitatory synapses elimination occurring during late adolescence in rats. The structural plasticity impact is associated with electrophysiological differences such as an increased excitability of the D2R +/- neurons as well as an increase in the miniature EPSC frequency. The authors then hypothesized that the effect may be linked to changes in LTD mechanisms and using TFS in the layer 2/3 and measure of the Peak amplitude in patched layer5 pyramidal neurons. Remarkably, the spine density and electrophysiological effects induced by D2R downregulation are mimicked using Eticlopride, a D2R antagonist. Finally, injection of this drug in the ACC induces anxiety-like behaviors.

Strengths:

I deeply appreciate the quality of the experiments and the large panel of methods used to support the main messages of the paper. The specific time window of the effect of the d2r downregulation is really interesting at the fundamental level but also for a better understanding of psychiatric diseases.

The results are well illustrated and explained and the stats are adequate with a good sample size for neurons and animals. However, a few points need to be corrected for this article to be published on Nature communications.

Response – We thank this reviewer for his/her comments that “I deeply appreciate the quality of the experiments...”, “The specific time window of the effect of D2R downregulation is really interesting...”, and “The results were well illustrated and explained...”. We are also grateful for the constructive critiques which will significantly improve the manuscript.

Weaknesses and points to correct:

In Methods:

1) The pixel sizes in microns for X-Y are missing.

Response – The pixel sizes in microns for X-Y are 1024×1024 px. We added this information in the revised manuscript (line 556-557).

2) The authors need to better explain their methods of classification for the mushroom, thin and stubby spines and give in supplementary figures different examples of each category.

Response – We clarified the methods of classification for the different types of spines (line 558-560) and showed the examples of each category (Supplementary Fig. 4) in the revised manuscript.

3) In the case of subjective classification, it is required to show as well objective spine measurements: length and spine head size, to be transparent and for other labs to reproduce it.

Response – We showed the length and spine head size for objective spine measurements, as suggested, in the revised manuscript (line 558-560 and Supplementary Fig. 4).

In Results:

4) The idea that interneurons express less D2R than pyramidal cells seems a bit over-interpreted. Interneurons represent on average around 20% of the cortical neurons, it makes sense to me that D2+ interneurons represent only 9/10% of the total, which is still almost half of the interneurons. The authors chose to study pyramidal cells and it is fine, but telling that D2R is mainly expressed in pyramidal cells is misleading considering the importance of interneurons for psychiatric disorders and especially for schizophrenia. Also, interneurons are major regulators of neuronal network activity, so the effects on behaviors might still be dependent on them. I propose that the authors adapt their justification of studying pyramidal neurons to be more objective.

Response – Sorry for the over-interpretation. We agree with this reviewer that Drd2⁺ interneurons are almost half of the total interneurons (Fig. 1g) and GABAergic interneurons are important for neural network activity. Here we focused on the pruning of dendritic spines which are mainly from pyramidal neurons. We stated as such and adjusted the reasons to study pyramidal neurons in the revised manuscript (line 154-158, 170-171).

5) The authors have a really good genetic model of D2R downregulation, so, using Eticlopride for behavioral studies, seems a bit surprising. It would be interesting to know if the behavioral effects of genetic D2R downregulation are the same as the effects of the drug.

Response – Sorry for being unclear. The Drd2^{+/-} rats have reduction in Drd2 expression globally, which made the behavioral results difficult to be interpreted. By contrast, the treatment with Drd2 antagonist Eticlopride was specific for the ACC brain regions. The behavioral phenotype of Drd2^{+/-} rats should be different from the rats receiving ACC injection of Eticlopride. For example, the Drd2^{+/-} rats were hypoactive in the open field (data not shown) while ACC injection of Eticlopride did not affect locomotion in the open field. We discussed this point in the revised manuscript (line 451-453).

6) In figure 2.d the GAPDH is overexposed, an exposure similar to 2.c would allow a proper quantification.

Response – The overexposed image was replaced with a more representative one (Fig. 2d).

7) *In figure 2,3,4 some of the dendrites showed in examples are blurry, probably due to a movement of the samples during the z-stack. I would like the authors to find better illustrative neurons*

Response – The blurry dendrites were replaced with more clear or sharpened images.

Discussion:

8) *In the 2nd paragraph, “Drd2 increased dendritic spines in hippocampal pyramidal”, a term is missing: “density”*

Response – The word of “density” was added (line 409).

9) *There is still a chance that behavioral effects are due to changes in presynaptic dopamine release through presynaptic D2R projecting into the ACC, independently of the synaptic pruning, this aspect can be discussed as well.*

Response – Agree. We discussed the potential roles of presynaptic Drd2 in behavioral deficits in the revised manuscript (line 453-455).

10) *D2+/- neurons are hyperexcitable and the authors showed it nicely, however, this effect might also be dependent on intrinsic excitability due to changes in ion channel properties as well, for example, due to cAMP and HCN1 or even due to effects on Voltage-gated channels. The authors could discuss those aspects as well.*

Response – Agree. We discussed the effects of Drd2^{+/-} on intrinsic excitability and ion channels in the revised manuscript (line 438-441).

Reviewer #3 (Remarks to the Author):

In their paper entitled “Dopamine D2 receptor regulates cortical synaptic pruning” Ya-Qiang Zhang et al., investigated the role of the dopamine receptor 2 (Drd2) in synaptic pruning during adolescence in a self-reporting Drd2 heterozygous rat. They found that Drd2 may play important roles in synaptic pruning during the transition from adolescence to adulthood without effects on spine formation. They demonstrated that the role of Drd2 in NMDAR-mediated LTD is cell-autonomous and that inhibition of Drd2 during adolescence may increase glutamatergic transmission in adults. This has important implications in synaptic plasticity at crucial developmental stages, and thus they highlight the potential role of Drd2 in development of anxiety.

The flow of the paper was good, and I want to compliment the diagrams in the figures that offer a visual explanation of the experiments performed.

However, there are some comments that I would like the authors to address before

recommending the paper for publication.

Response – We thank this reviewer for his/her comments that “The flow of the paper was good, and I want to compliment the diagrams in the figures...”. We are also grateful for the constructive critiques which will significantly improve the manuscript.

Minor comments:

- I suggest that authors adding the age of the rats when referred to periods like “adolescence” and “adulthood” to remind the reader the time points used for each developmental stage (for example, in line 160, line 304, etc...)

Response – Good suggestion. We added the age of the rats when referred to periods like “adolescence” and adulthood” in the revised manuscript (line 354, 356, 361-364, 376-378, 383, 385).

- In line 201 stubby and mushroom-shaped spines are grouped as mature spines. Other researchers have considered stubby morphologies as immature or transitory, rather than mature (see references below). Why do authors consider stubby morphologies mature in this work? Can the authors provide other references that indicate this classification as accepted?*

**See:*

Harris KM, Jensen FE, Tsao B. Three-dimensional structure of dendritic spines and synapses in rat hippocampus (CA1) at postnatal day 15 and adult ages: implications for the maturation of synaptic physiology and long-term potentiation. J Neurosci. 1992 Jul;12(7):2685-705. doi: 10.1523/JNEUROSCI.12-07-02685.1992. PMID: 1613552; PMCID: PMC6575840.

Berry KP, Nedivi E. Spine Dynamics: Are They All the Same? Neuron. 2017 Sep 27;96(1):43-55. doi: 10.1016/j.neuron.2017.08.008. PMID: 28957675; PMCID: PMC5661952.

Pchitskaya E, Bezprozvanny I. Dendritic Spines Shape Analysis-Classification or Clusterization? Perspective. Front Synaptic Neurosci. 2020 Sep 30;12:31. doi: 10.3389/fnsyn.2020.00031. PMID: 33117142; PMCID: PMC7561369

Response – Sorry for the confusion. We agree with this reviewer that only the mushroom-like spines represent mature spines. We revised the text and stated as such (line 214-216).

- In line 243 authors mention that Drd2 is important in NMDAR-mediated LTD. Other groups have looked into this and should be cited. (e.g. Sheynikhovich et al., J Neuroscience 2013).

Response – We cited the paper in the revised manuscript (line 312).

- I suggest to include the age of the rats used for each experiment in the methods section to have all the information handy for the reader.

Response – We included the age of the rats used for each experiment in the revised method section (line 486-492), as suggested.

Major comments:

- Authors should explain the criteria used to classify spines in the different categories (mushroom, thin and stubby). If this classification was subjective to the investigator analyzing the images, I strongly suggest to repeat the classification using an unbiased method. There is available (cost-free) software such as NeuronStudio software (Computational Neurobiology and Imaging Center CNIC, Mount Sinai School of Medicine, NY RRID:SCR_005793) or Reconstruct (Fiala, J.C. (2005), Reconstruct: a free editor for serial section microscopy. Journal of Microscopy, 218: 52-61. <https://doi.org/10.1111/j.1365-2818.2005.01466.x>).

Response – Good suggestion. We used the Reconstruct software to reanalyze the different types of spines (Fig. 2i and 7g). We added the detailed methods on how to characterize the different categories of spines in the revised manuscript (line 558-560).

- In the Statistical Analysis of the Methods section the authors quote the Wikipedia as the source to decide on their approach to eliminate outliers from their datasets. The authors don't explain properly why they decided that eliminating outliers was necessary. Outliers should be calculated with a proper statistical test such as Grubbs's test. If a big section of the datasets were eliminated through their formula, the reality may be that there are no significant effects. If the biological variability was too wide, normalizing data would be more appropriate than eliminating data points. Authors should provide complete datasets including the outliers that have been removed, and explain why they deemed necessary to calculate and eliminate outliers.

Response – Good suggestion. We calculated the outliers with the Grubb's test, as suggested. We stated as such in the revised method (line 702-703). Actually, the number of outliers is only one, as indicated by the red color in the source data (one data point in Fig. 9g). All the data except one outlier was included in the statistical analysis.

Reference

Beaulieu, J.M., Sotnikova, T.D., Marion, S., Lefkowitz, R.J., Gainetdinov, R.R., and Caron, M.G. (2005). An Akt/beta-arrestin 2/PP2A signaling complex mediates dopaminergic neurotransmission and behavior. *Cell* 122, 261-273.

Beaulieu, J.M., Tirotta, E., Sotnikova, T.D., Masri, B., Salahpour, A., Gainetdinov, R.R., Borrelli, E., and Caron, M.G. (2007). Regulation of Akt signaling by D2 and D3

dopamine receptors in vivo. *J Neurosci* 27, 881-885.

Bonito-Oliva, A., Pallottino, S., Bertran-Gonzalez, J., Girault, J.A., Valjent, E., and Fisone, G. (2013). Haloperidol promotes mTORC1-dependent phosphorylation of ribosomal protein S6 via dopamine- and cAMP-regulated phosphoprotein of 32 kDa and inhibition of protein phosphatase-1. *Neuropharmacology* 72, 197-203.

Keshavan, M.S., Anderson, S., and Pettegrew, J.W. (1994). Is schizophrenia due to excessive synaptic pruning in the prefrontal cortex? The Feinberg hypothesis revisited. *J Psychiatr Res* 28, 239-265.

Konopaske, G.T., Lange, N., Coyle, J.T., and Benes, F.M. (2014). Prefrontal cortical dendritic spine pathology in schizophrenia and bipolar disorder. *JAMA Psychiatry* 71, 1323-1331.

Lee, C.M., Sylantsev, S., and Shyu, B.C. (2007). Short-term synaptic plasticity in layer II/III of the rat anterior cingulate cortex. *Brain Res Bull* 71, 416-427.

Penzes, P., Cahill, M.E., Jones, K.A., VanLeeuwen, J.E., and Woolfrey, K.M. (2011). Dendritic spine pathology in neuropsychiatric disorders. *Nat Neurosci* 14, 285-293.

Puighermanal, E., Biever, A., Espallergues, J., Gangarossa, G., De Bundel, D., and Valjent, E. (2015). *drd2-cre:ribotag* mouse line unravels the possible diversity of dopamine d2 receptor-expressing cells of the dorsal mouse hippocampus. *Hippocampus* 25, 858-875.

Rocchetti, J., Isingrini, E., Dal Bo, G., Sagheby, S., Menegaux, A., Tronche, F., Levesque, D., Moquin, L., Gratton, A., Wong, T.P., *et al.* (2015). Presynaptic D2 dopamine receptors control long-term depression expression and memory processes in the temporal hippocampus. *Biol Psychiatry* 77, 513-525.

Tang, G., Gudsnuk, K., Kuo, S.H., Cotrina, M.L., Rosoklija, G., Sosunov, A., Sonders, M.S., Kanter, E., Castagna, C., Yamamoto, A., *et al.* (2014). Loss of mTOR-dependent macroautophagy causes autistic-like synaptic pruning deficits. *Neuron* 83, 1131-1143.

Yu, Q., Liu, Y.Z., Zhu, Y.B., Wang, Y.Y., Li, Q., and Yin, D.M. (2019). Genetic labeling reveals temporal and spatial expression pattern of D2 dopamine receptor in rat forebrain. *Brain Struct Funct* 224, 1035-1049.

Reviewers' Comments:

Reviewer #1:

Remarks to the Author:

We thank the authors for careful consideration of our critiques and significant additional experimental work that we think makes the manuscript much stronger. The study provides the field with important data on developmental changes in ACC synapses and points us towards specific D2-mediated mechanisms for developmental spine pruning in these cells. We especially appreciate the new experiments showing rapamycin rescue of the spine density phenotype and targeted knockdown experiments that much better make the case for a cell-intrinsic mechanism. There are a few minor points remaining.

Minor points:

ll.111-112; 127-128; 300-301; 395-396: The replacement language that we requested in these passages remains grammatically confusing (i.e. should be "includes" rather than "included"; "similar...processes as" rather than "similar...with"; etc.)

ll.234-236: We still think that the language here conflates the changes in excitability implied by spine differences and those revealed by whole cell responses to current injection. Something like "In addition to differences in dendritic spine densities, we tested whether reduced D2R expression mediated cell-intrinsic excitability as revealed by responses to direct depolarizing current injections" may be less confusing than trying to say that one result implies the other.

ll.248: should be " β -arrestin" (not "arretin")

ll.272: "Drd2-positive"

ll.428-431: We recommend removing the statement about lack of dopamine cells in these slices. While the slices don't have dopamine cell bodies, dopamine axons are functional and still release neurotransmitter in coronal slices. The differences in expression that are subsequently cited are sufficient to make the point.

Reviewer #2:

Remarks to the Author:

We can appreciate the quality of the experiments realized as a response to the first review. The publication is significantly improved. However, a few changes will be required before publication. Figure 7.d. Veh 4W in fig 7. d. is blurry, which is an issue for visualizing spine morphology, spines look duplicated because of that, the authors should select a clearer image.

Methods. 1- Rapamycin is usually diluted in DMSO, does the vehicle contain it as well? Clarification on this would be needed as the methods only mentioned saline as vehicle.

2- Details of the pAAV-EF1 α -loxP-stop-loxP-tdTomato-WPRE-poly serotypes are needed for better reproducibility in the future (for example AAV1, 9 etc)

3- For the "miRNA", are the authors meaning shRNA by any chance? Even though miRNA can reduce expression, they reduce expression of several genes and their use to target a single gene is rare.

Typo: Drd2-positie on line 274, a "v" is missing in positive

Reviewer #3:

Remarks to the Author:

I appreciate the effort that the authors put into addressing my previous comments in a satisfactory manner.

I would like the authors to address the following minor comments which I believe will improve the

flow of the manuscript. Once these minor points are fixed, I think the manuscript meets Nature Communications standards.

In line 214 authors make the conclusion regarding the mature spines, after agreeing with me and previous reports by other groups, that only mushroom spines are considered mature. However, authors also see a similar (and also significant) increase in the stubby morphologies, but they don't comment on these results. What is the meaning of this increase in the opinion of the authors? This may mean that, along with increase of mature spines, there is a possibility that Drd2 is also involved with spinogenesis. I think this is something that authors should acknowledge and make their interpretations.

In line 558 authors addressed my comment about using an unbiased method to classify spine morphology. They used Reconstruct software to classify spines. Did they also use it for counting? A few lines later (562-564) they say they classify spines manually. Please, clarify which method was used for quantification, if the same was used for counting, and which results are reflected in the graphs in figure 2. Could authors also explain in a bit more detail the criteria used? For example: Is 0.6µm referring to the diameter of the head size, circumference...?, what do authors mean by "ratio"?

Line 245-297, 451-455 need some polishing. The writing can be improved for clarity, and typos and grammar mistakes need to be corrected in general throughout the entire manuscript (newly added lines and sections).

One of the reviewers suggested to change SR-Drd2+/- to simply Drd2+/- . However, the authors are now mixing both nomenclatures throughout the paper which I find confusing. I suggest they decide on one nomenclature, explain it at the beginning, and stick with it for the rest of the manuscript (including figures and legends).

Throughout the discussion section, it would be helpful for the reader if authors reference the corresponding figure where the results they are discussing are presented.

Line 439, do they refer to Drd2-positive neurons or Drd2-negative neurons?

Line 508: details of the imaging (objective, NA, pixel size, etc...) are missing.

Line 556: The authors state the pixel size in microns but they provide the total pixel size of the image (1024 x 1024) which was already mentioned in the previous line.

Reviewer #1 (Remarks to the Author):

We thank the authors for careful consideration of our critiques and significant additional experimental work that we think makes the manuscript much stronger. The study provides the field with important data on developmental changes in ACC synapses and points us towards specific D2-mediated mechanisms for developmental spine pruning in these cells. We especially appreciate the new experiments showing rapamycin rescue of the spine density phenotype and targeted knockdown experiments that much better make the case for a cell-intrinsic mechanism. There are a few minor points remaining.

Minor points:

ll.111-112; 127-128; 300-301; 395-396: The replacement language that we requested in these passages remains grammatically confusing (i.e. should be "includes" rather than "included"; "similar...processes as" rather than "similar...with"; etc.)

Response – We revised the sentence as suggested: which includes some similar cellular/biochemical processes as synaptic pruning.

ll. 234-236: We still think that the language here conflates the changes in excitability implied by spine differences and those revealed by whole cell responses to current injection. Something like "In addition to differences in dendritic spine densities, we tested whether reduced D2R expression mediated cell-intrinsic excitability as revealed by responses to direct depolarizing current injections" may be less confusing than trying to say that one result implies the other.

Response – We revised the sentence as suggested: In addition to differences in dendritic spine densities, we tested whether reduced *Drd2* expression affected cell-intrinsic excitability as revealed by responses to direct depolarizing current injections.

ll.248: should be " β -arrestin" (not "arretin")

Response – We corrected the typos.

ll.272: "Drd2-positive"

Response – We corrected the typos.

ll.428-431: We recommend removing the statement about lack of dopamine cells in these slices. While the slices don't have dopamine cell bodies, dopamine axons are functional and still release neurotransmitter in coronal slices. The differences in expression that are subsequently cited are sufficient to make the point.

Response – We removed the statement about lack of dopamine cells in the ACC slices, as suggested.

Reviewer #2 (Remarks to the Author):

We can appreciate the quality of the experiments realized as a response to the first review. The publication is significantly improved. However, a few changes will be required before publication.

Figure 7.d. Veh 4W in fig 7. d. is blurry, which is an issue for visualizing spine morphology, spines look duplicated because of that, the authors should select a clearer image.

Response – We replace the blurry image with a clear one in Fig. 7d, as suggested.

Methods. 1- Rapamycin is usually diluted in DMSO, does the vehicle contain it as well? Clarification on this would be needed as the methods only mentioned saline as vehicle.

Response – DMSO was used as the vehicle control in the experiments with rapamycin. We clarified this point in the revised method.

2- Details of the pAAV-EF1 α -loxP-stop-loxP-tdTomato-WPRE-poly serotypes are needed for better reproducibility in the future (for example AAV1, 9 etc)

Response – The serotype of AAV used in this study is AAV2/9, composed by the AAV2 rep and AAV9 cap genes. We added this information in the revised method.

3- For the “miRNA”, are the authors meaning shRNA by any chance? Even though miRNA can reduce expression, they reduce expression of several genes and their use to target a single gene is rare.

Response – Sorry for being confusing. Here we used an shRNA expressing system called PRIMER (potent RNA interference using microRNA expressing vectors)¹, which allows for the multicistronic cotranscription of a reporter gene, thereby facilitating the tracking of shRNA production in individual cells. It is *Drd2* shRNA that downregulate *Drd2* expression. We just put the shRNA in the microRNA expressing vector. We clarified this issue in the revised method.

Typo: Drd2-positie on line 274, a “v” is missing in positive

Response – We corrected the typos.

Reviewer #3 (Remarks to the Author):

I appreciate the effort that the authors put into addressing my previous comments in a satisfactory manner. I would like the authors to address the following minor comments which I believe will improve the flow of the manuscript. Once these minor points are fixed, I think the manuscript meets Nature Communications standards.

*In line 214 authors make the conclusion regarding the mature spines, after agreeing with me and previous reports by other groups, that only mushroom spines are considered mature. However, authors also see a similar (and also significant) increase in the stubby morphologies, but they don't comment on these results. What is the meaning of this increase in the opinion of the authors? This may mean that, along with increase of mature spines, there is a possibility that *Drd2* is also involved with spinogenesis. I think this is something that authors should acknowledge and make their interpretations.*

Response – We agree with this reviewer and stated as such: the increase of stubby spines might suggest that *Drd2* is also involved with spinogenesis.

In line 558 authors addressed my comment about using an unbiased method to classify spine morphology. They used Reconstruct software to classify spines. Did they also use it for counting? A few lines later (562-564) they say they classify spines manually. Please, clarify which method was used for quantification, if the same was used for counting, and which results are reflected in the graphs in figure 2. Could authors also explain in a bit more detail the criteria used? For example: Is 0.6µm referring to the diameter of the head size, circumference...?, what do authors mean by "ratio"?

Response – We used Reconstruct software for both quantification and counting (Fig. 2g-i). We forgot to delete the sentence of manual counting. Sorry for the oversight. The 0.6 µm is referring to the diameter of the spine head. The ratio means the spine length divided by the width of spine base. We clarified these points in the revised method.

Line 245-297, 451-455 need some polishing. The writing can be improved for clarity, and typos and grammar mistakes need to be corrected in general throughout the entire manuscript (newly added lines and sections).

Response – We improved the clarity and corrected the typos and grammar mistakes, as suggested.

*One of the reviewers suggested to change *SR-Drd2*+/- to simply *Drd2*+/-. However, the authors are now mixing both nomenclatures throughout the paper which I find*

confusing. I suggest they decide on one nomenclature, explain it at the beginning, and stick with it for the rest of the manuscript (including figures and legends).

Response – We stick to use SR-*Drd2*^{+/-} because it has different genotypes from *Drd2*^{+/-}.

Throughout the discussion section, it would be helpful for the reader if authors reference the corresponding figure where the results they are discussing are presented.

Response – We referenced the corresponding figure where the results were discussed in the revised manuscript.

Line 439, do they refer to Drd2-positive neurons or Drd2-negative neurons?

Response – We refer to *Drd2*-positive neurons, which is clarified in the revised manuscript.

Line 508: details of the imaging (objective, NA, pixel size, etc...) are missing.

Response – We provided the details of imaging in the revised manuscript.

Line 556: The authors state the pixel size in microns but they provide the total pixel size of the image (1024 x 1024) which was already mentioned in the previous line.

Response – The pixel sizes in microns for x-y are 0.06. We clarified this issue in the revised manuscript.

Reference:

1. Stegmeier, F., Hu, G., Rickles, R.J., Hannon, G.J. & Elledge, S.J. A lentiviral microRNA-based system for single-copy polymerase II-regulated RNA interference in mammalian cells. *Proc Natl Acad Sci U S A* **102**, 13212-13217 (2005).